# Convergent mosaic brain evolution is associated with the evolution of novel electrosensory systems in teleost fishes

Erika L Schumacher, Bruce A Carlson*

Department of Biology, Washington University, St Louis, United States

**Abstract** Brain region size generally scales allometrically with brain size, but mosaic shifts in brain region size independent of brain size have been found in several lineages and may be related to the evolution of behavioral novelty. African weakly electric fishes (Mormyroidea) evolved a mosaically enlarged cerebellum and hindbrain, yet the relationship to their behaviorally novel electrosensory system remains unclear. We addressed this by studying South American weakly electric fishes (Gymnotiformes) and weakly electric catfishes (*Synodontis* spp.), which evolved varying aspects of electrosensory systems, independent of mormyroids. If the mormyroid mosaic increases are related to evolving an electrosensory system, we should find similar mosaic shifts in gymnotiforms and *Synodontis*. Using micro-computed tomography scans, we quantified brain region scaling for multiple electrogenic, electroreceptive, and non-electrosensing species. We found mosaic increases in cerebellum in all three electrogenic lineages relative to non-electric lineages and mosaic increases in torus semicircularis and hindbrain associated with the evolution of electrogenesis and electroreceptor type. These results show that evolving novel electrosensory systems is repeatedly and independently associated with changes in the sizes of individual major brain regions independent of brain size, suggesting that selection can impact structural brain composition to favor specific regions involved in novel behaviors.

*For correspondence:
carlson.bruce@wustl.edu

**Competing interest:** The authors declare that no competing interests exist.

## Editor's evaluation

Much of the observed variation in brain region volumes across vertebrates is explained by the scaling relationship of each region with brain size. Nevertheless, mosaic shifts in region volumes independent of overall brain size appear and are thought to reflect selection on particular behavioral traits associated with those brain regions. The work reported here used the independently evolved electrosensory systems of African weakly electric fish, South American weakly electric fish, and weakly electric catfish to show similar enlargement of brain regions, suggesting that selection is repeatedly able to favor brain regions involved in specific behaviors.

## Introduction

Brains are composed of multiple regions that vary widely in size across vertebrates and are associated with particular functions and behaviors (*Striedter, 2005*; *Striedter and Northcutt, 2020*). Much of the variation in brain region sizes is attributed to the allometric scaling of each region with total brain size (concerted evolution), which may result from conservation and constraint in developmental neurogenesis (*Finlay and Darlington, 1995*; *Striedter, 2005*). Seemingly disproportionately enlarged regions can have larger allometric slopes by extending the timing of neurogenesis for late developing brain regions such as the cortex and cerebellum (*Finlay and Darlington, 1995*). However, changes in brain region sizes independent of total brain size, or mosaic shifts, have also been observed in several

**eLife digest** Larger animals tend to have larger brains and smaller animals tend to have smaller ones. However, some species do not fit the pattern that would be expected based on their body size. This variation between species can also apply to individual brain regions. This may be due to evolutionary forces shaping the brain when favouring particular behaviours. However, it is difficult to directly link changes in species behaviour and variations in brain structure.

One way to understand the impact of evolutionary adaptations is to study species that have developed new behaviours and compare them to related ones that lack such a behaviour. An opportunity to do this lies in the ability of several species of fish to produce and sense electric fields in water. While this system is not found in most fish, it has evolved multiple times independently in distantly-related lineages. Schumacher and Carlson examined whether differences in the size of brains and individual regions between species were associated with the evolution of electric field generation and sensing.

Micro-computed tomography, or µCT, scans of the brains of multiple fish species revealed that the species that can produce electricity – also known as 'electrogenic' species' – have more similar brain structures to each other than to their close relatives that lack this ability. The brain regions involved in producing and detecting electrical charges were larger in these electrogenic fish. This similarity was apparent despite variations in how total brain size has evolved with body size across species.

These results demonstrate how evolutionary forces acting on particular behaviours can lead to predictable changes in brain structure. Understanding how and why brains evolve will allow researchers to better predict how species' brains and behaviours may adapt as human activities alter their environments.

taxa and are hypothesized to reflect selection on traits associated with those regions (**Barton and Harvey, 2000**; **Striedter, 2005**). Mosaic shifts in fine-scale brain regions and circuits are well accepted and have been linked to changes in behavior (**Carlson et al., 2011**; **Vélez et al., 2017**; **Vélez et al., 2019**; **Gutiérrez-Ibáñez et al., 2014**; **Moore and DeVoogd, 2017**; **DeCasien and Higham, 2019**; **Krebs, 1990**), but the scale at which selection can act to alter brain region sizes remains unclear. There may potentially be more flexibility for mosaic changes in nuclei or circuits dedicated to specific functions compared to major brain regions that serve multiple functions and may be subject to greater developmental and phylogenetic constraints.

Most studies looking at the scaling of major brain regions instead find evidence of concerted evolution (**Finlay and Darlington, 1995**; **Striedter, 2005**; **Yopak et al., 2010**). There is some evidence of mosaic evolution at these scales (**Hoops et al., 2017**; **Sukhum et al., 2018**), but the drivers and selective pressures necessary for mosaic evolution to overcome constraints, whether developmental or due to functional interconnectivity, remain unclear. Further, this is difficult to test without repeated evolution of the same phenotypes.

Mosaic brain evolution of major brain regions is hypothesized to occur more frequently at larger taxonomic scales and alongside behavioral innovations that open new niches since mosaic shifts are more likely to contribute to major differences in brain function (**Striedter, 2005**). In dragon lizards, mosaic brain evolution is associated with ecomorph (species similar in morphology and behavior inhabiting the same ecological niche; **Hoops et al., 2017**), but as many different behavioral and sensory changes occur alongside ecomorph development, it is difficult to identify specific selective pressures favoring the observed ecomorph brain structure. Weakly electric fishes are excellent for testing whether mosaic brain evolution occurs with behavioral novelty: these fishes evolved behaviorally novel active electrosensory systems with several neural innovations, which likely resulted in strong selection for electrosensory processing capabilities (**Carlson and Arnegard, 2011**). Further, multiple lineages independently evolved similar electrosensory systems (**Crampton, 2019**).

Previous studies found that African weakly electric fishes (Mormyroidea) evolved extremely large brains along with mosaic increases in the sizes of the cerebellum and hindbrain relative to other non-electric osteoglossiforms (**Sukhum et al., 2018**; **Sukhum et al., 2016**). These mosaic increases occurred alongside the evolution of an active electrosensory system (electrogenesis + electroreception), but since this is only a single lineage, it is impossible to determine whether these mosaic shifts are associated with the evolution of this electrosensory system or with other phenotypes that

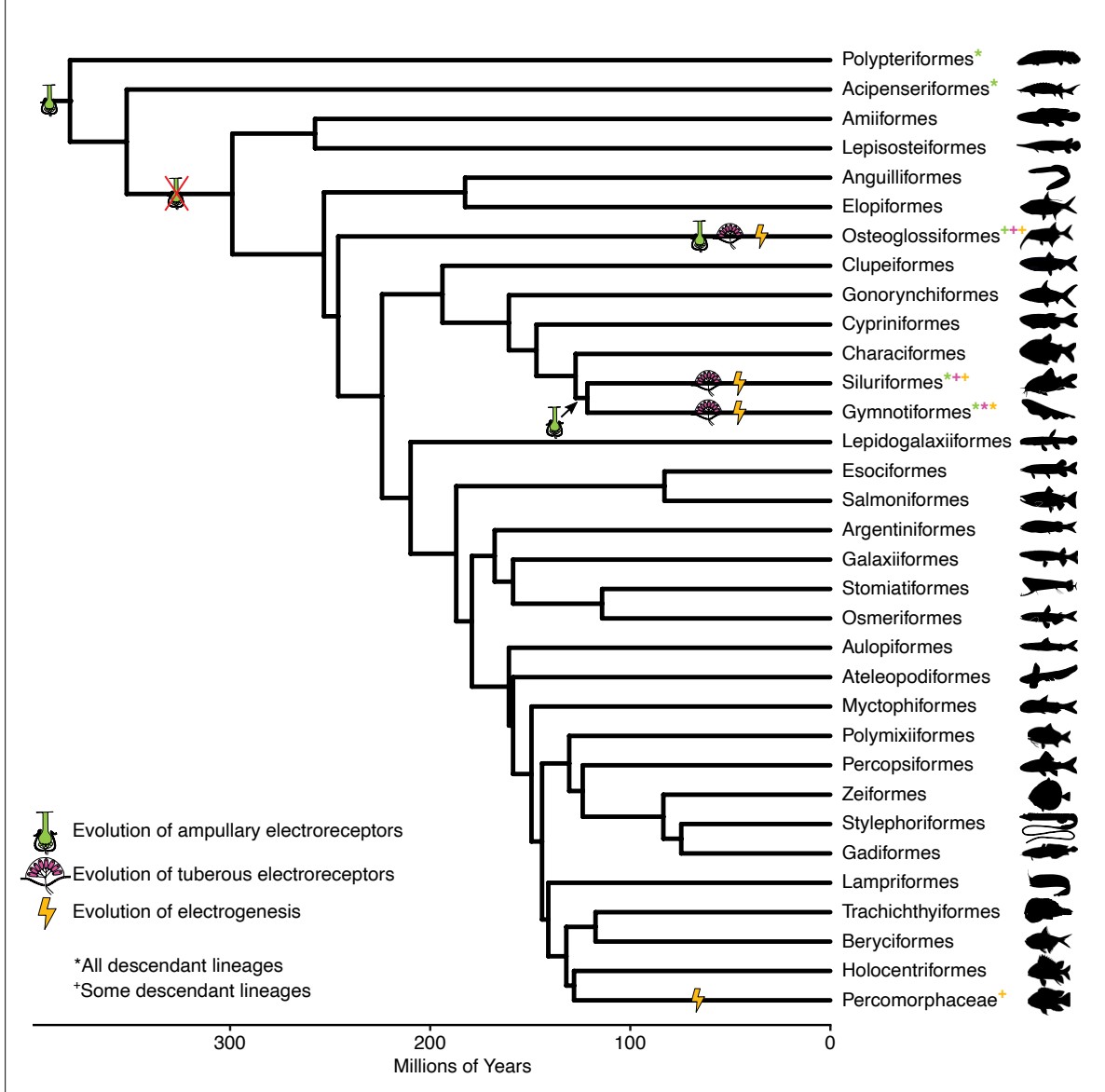

**Figure 1.** Chronogram of ray-finned fish orders, based on *Hughes et al., 2018*, showing the evolution of electrosensory phenotypes. Symbols indicate independent origins of each electrosensory phenotype. * indicates all descendant lineages have that electrosensory phenotype while + indicates some descendant lineages have that electrosensory phenotype. Green, ampullary electroreceptors; magenta, tuberous electroreceptors; orange, electrogenesis.

differentiate mormyroids from their closest living relatives. Further, one non-electric osteoglossiform, *Xenomystus nigri*, is electroreceptive, and there is no evidence that it has experienced mosaic brain evolution compared to other non-electric osteoglossiforms.

Electrogenesis and electroreception have evolved multiple times, but sparingly, across vertebrates—at least six times and at least two times, respectively, within ray-finned fishes (Actinopterygii; *Figure 1*). However, the degree to which electrosensory systems have evolved, with respect to their component parts and how they are utilized, varies across these different independent origins. For example, some lineages produce strong electrical discharges while others produce weak electrical discharges, some lineages have evolved multiple types of electroreceptors while others have evolved just one, and the usage and developmental origins of electrogenesis differ across lineages (*Crampton, 2019*). Here, we

investigated another lineage of fishes, otophysans, which includes taxa that evolved similar electrosensory systems independent of mormyroids, but to varying degrees with respect to their components and utilization, to determine whether these mosaic shifts are found repeatedly alongside the evolution of active electrosensing. Although osteoglossiform and otophysan lineages have convergently evolved similar electrosensory systems (*Crampton, 2019*), there are some distinctive differences in the degree of electrosensory usage, electroreceptor type, and electrical discharge type that could indicate differential selective pressures on the brains of these species. These differences allowed us to assess how multiple aspects of electrosensory systems relate to mosaic brain evolution.

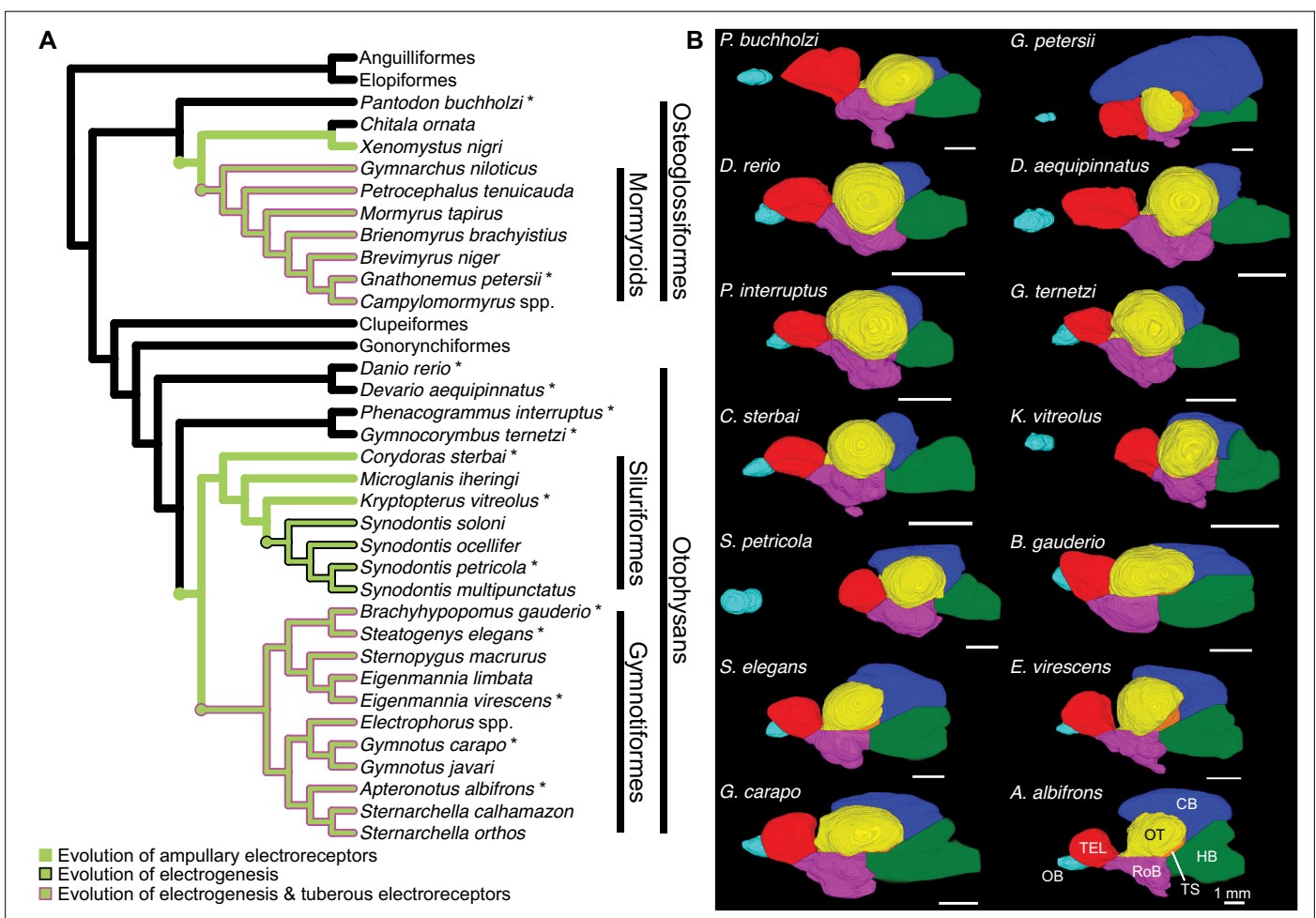

**Figure 2.** Brain morphology varies across species. (**A**) Cladogram of the inferred phylogenetic relationships of species included in this study (N = 32) and the orders between them. Order level relationships are based on *Hughes et al., 2018*. Green branches represent presence of ampullary electroreceptors. Black outline represents electrogenic species while the magenta outline represents electrogenic species with tuberous electroreceptors. (**B**) Example 3D reconstructions of brains from this study; these species are indicated on the cladogram with an asterisk. Brains are oriented from a lateral view with anterior to the left and dorsal at the top. Brain regions are color coded: OB, olfactory bulbs (cyan); TEL, telencephalon (red); HB, hindbrain (green); OT, optic tectum (yellow); TS, torus semicircularis (orange); CB, cerebellum (blue); RoB, rest of brain (magenta). Scale bar = 1 mm.

The online version of this article includes the following video and figure supplement(s) for figure 2:

**Figure supplement 1.** Electric discharges recorded from *Synodontis* spp.

**Figure 2—video 1.** Example 3D reconstructions of brains from Figure 1B.

https://elifesciences.org/articles/74159/figures#fig2video1

## Results

To investigate how electrosensory systems relate to structural brain variation, we combined published osteoglossiform data for electrogenic mormyroids, electroreceptive *Xenomystus*, and non-electrosensory outgroup species (*Sukhum et al., 2018*) with otophysan data for two additional electrogenic lineages (Gymnotiformes and *Synodontis* Siluriformes), electroreceptive (but not electrogenic) siluriforms, and non-electrosensory Characiformes and Cypriniformes (outgroup otophysans). Using micro-computed tomography (μCT) scans, we measured total brain and brain region volumes for 15 electrogenic, 3 electroreceptive, and 4 non-electrosensory otophysan species. Combined with the published osteoglossiform data, this yielded a dataset of 32 species (*Figure 2*, *Figure 2—video 1*). We measured the volumes of seven distinct brain regions (olfactory bulbs, OB; telencephalon, TEL; hindbrain, HB; optic tectum, OT; torus semicircularis, TS; cerebellum, CB; and rest of brain, RoB) to determine patterns of major brain region scaling across taxa. Rest of brain includes thalamus, hypothalamus, and additional midbrain structures excluding optic tectum and torus semicircularis (see 'Brain region delimitation').

### Electrosensory system evolution is not associated with shifts in total brain size scaling

A recent study found a steeper brain–body allometric relationship for osteoglossiforms compared to other actinopterygians, but not for seven other focal ray-finned fish orders, which may, in part, be driven by the highly speciose mormyroids (*Tsuboi, 2021*). To determine if the evolution of an electrosensory system is associated with extreme encephalization or shifts in brain–body allometric relationships, we combined our data with published brain and body mass data across ray-finned fishes (*Tsuboi, 2021*; *Tsuboi et al., 2018*), which resulted in a combined dataset of 870 species across 46 orders, with phylogenetic data from a previously assembled time-calibrated phylogeny (*Rabosky et al., 2018*). We used Bayesian reversible-jump bivariate multiregime Ornstein–Uhlenbeck modeling (OUrjMCMC; *Uyeda et al., 2017*) to identify shifts in both y-intercept and slope of brain–body allometric relationships. This approach allows shifts to be identified without assuming their location a priori.

We identified eight allometric shifts across actinopterygians, three in lineages with at least three descendants: osteoglossiforms, and two shifts within percomorphs (*Figure 3A*). The first (hereafter referred to as percomorph grade A) included Lophiiformes, Tetraodontiforms, Acanthuriformes, some descendants from Scorpaeniformes, and some descendants from Perciformes. The second (hereafter referred to as percomorph grade B) included Gasterosteiformes, some descendants from Scorpaeniformes, and some descendants from Perciformes. Shifts were also detected for *Gymnothorax meleagris, Synodontis multipunctatus, Arothron nigropunctatus,* and a mormyroid lineage containing *Isichthys henryi + Brienomyrus brachyistius*. All other taxa best fit the ancestral allometry. However, it is worthwhile to note that additional shifts may be present within actinopterygians. There are more than 20,000 known actinopterygians while we only have data for 870 species. Further, multiple single species were identified as having increased brain–body allometries; however, it remains unclear if these single species have particularly enlarged brains compared to their closest relatives or if a more speciose lineage with enlarged brains would be identified with additional sampling of close relatives.

For shifts in lineages (i.e., for each grade) containing at least three descendants, we tested the putative grades in a phylogenetic generalized least squares (PGLS) framework for shifts in both slope and y-intercept using a phylogenetically corrected analysis of covariance (ANCOVA) and phylogenetically corrected pairwise post-hoc testing with a Bonferroni correction. We found that all of the grades differ significantly in slope ($p<0.05$) while only osteoglossiforms relative to the ancestral grade and relative to percomorph grades A and B differ in y-intercept ($p<0.05$; *Figure 3B*, *Figure 3—source data 2*). However, only osteoglossiforms had an increased slope relative to the ancestral grade while the slope for percomorph grade A decreased relative to the ancestral grade with a further decrease in percomorph grade B. To confirm shifts across all putative grades, we fit the following PGLS models: all the identified shifts, each grade collapsed in turn to its ancestral grade, and a single allometric relationship across all taxa. Collapsing *S. petricola* to the ancestral grade was the best-fit model (ΔAIC > 3) with the model containing all putative shifts as second best (ΔAIC > 6; *Figure 3—source data 3*).

To explicitly address whether shifts in the allometric relationship are associated with the evolution of electrosensory phenotypes, we ran OUrjMCMC models with fixed shifts for lineages that evolved ampullary electroreceptors, lineages that evolved tuberous electroreceptors, and lineages

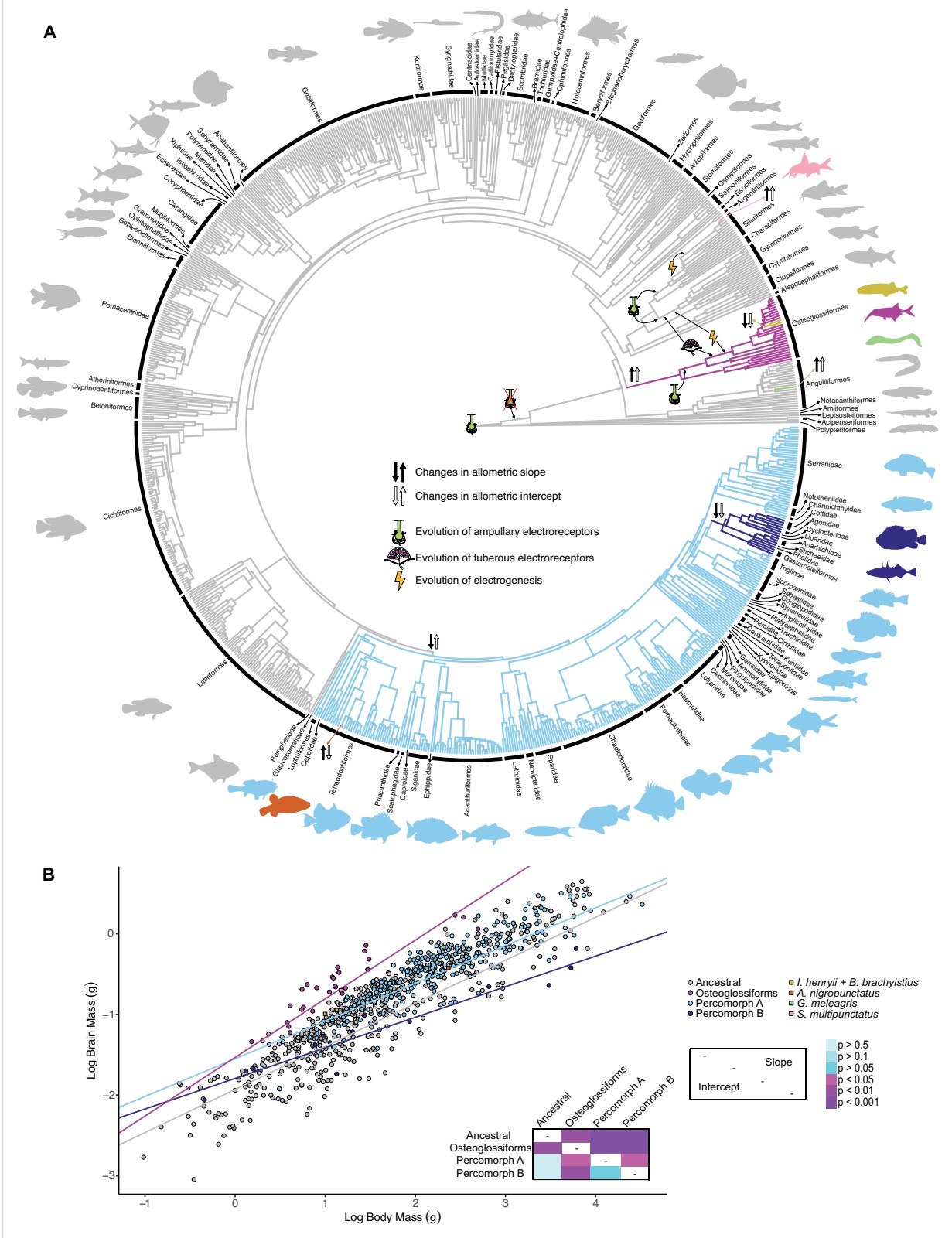

**Figure 3.** Mormyroids are more encephalized than gymnotiforms. (**A**) Chronogram of ray-finned fishes based on *Rabosky et al., 2018* showing shifts in the brain–body allometric relationship. Different branch colors indicate different allometric relationships. Direction of slope changes is indicated by black arrows, and direction of intercept changes is indicated by white arrows. Electrosensory phenotypes are indicated by symbols. Tree topology differs from the phylogenies in *Figures 1–2*, which highlights the well-established difficulty and discrepancies in resolving taxonomic relationships of ray-

*Figure 3 continued on next page*

*Figure 3 continued*

finned fishes. (**B**) Plot of log brain size by log body size. Points correspond to species means and are colored according to the identified grades in (A). Phylogenetic generalized least squares (PGLS) lines correspond to the distinct allometric relationships indicated in (**A**) and were determined for grades with at least three descendants. Inset shows a heatmap of the phylogenetically corrected pairwise post-hoc analysis of covariance (ANCOVA) results with a Bonferroni correction for differences in intercept below the diagonal and differences in slope above the diagonal. Significant differences are in shades of magenta/purple and nonsignificant differences are in shades of blue.

The online version of this article includes the following source data for figure 3:

**Source data 1.** Ornstein–Uhlenbeck modeling (OUrjMCMC) and phylogenetic generalized least squares (PGLS) fitted brain–body allometries for each grade.

**Source data 2.** Results (p-values) of phylogenetically corrected pairwise post-hoc tests with a Bonferroni correction for an analysis of covariance (ANCOVA) comparing phylogenetic generalized least squares (PGLS) relationships of brain size against body for each identified grade.

**Source data 3.** Model selection results for phylogenetic generalized least squares (PGLS) relationships systematically collapsing each putative grade to its ancestral grade.

**Source data 4.** Ornstein–Uhlenbeck modeling (OUrjMCMC) estimated marginal likelihoods and model selection results for shifts in brain–body allometries associated with electrosensory phenotypes.

that evolved electrogenesis in addition to the following null hypotheses: a fixed shift at the branch leading to osteoglossiforms as found previously (*Tsuboi, 2021*), only shifts in the allometric relationship for intercept but not slope, and a single allometric relationship across all taxa. We found that the model with eight shifts provided the best fit to the data (2ln(BF)>28; *Figure 3—source data 4*). In addition, when forcing a global slope across all taxa, we found no shifts in intercept with a posterior probability >0.1 despite all parameters having estimated sample sizes >1000. Taken together, these results suggest that fishes can evolve an active electrosensory system without evolving a brain as large as that of mormyroids.

## Electrogenic species have similar structural brain variation

To determine how brain structure varies in association with electrosensory phenotype, we used regional measurements for all taxa and ran a phylogenetically corrected principal components analysis (pPCA). Considering the phylogenetic relationships among otophysans are still debated (*Crampton, 2019*; *Hughes et al., 2018*; *Rabosky et al., 2013*), our goal was only to account for relatedness to the best of our ability, not to propose a resolved phylogeny. We found that electrogenic species cluster distinctly from both electroreceptive and non-electrosensory species, which overlap considerably (*Figure 4A*). All electroreceptive lineages (mormyroids, *Xenomystus*, gymnotiforms, and siluriforms) have evolved ampullary electroreceptors, which detect relatively low-frequency electrical information, while only gymnotiforms and mormyroids have evolved additional tuberous electroreceptors that broaden the frequency range of detectable signals (Crampton, 2019). We find that electrogenic species with both electroreceptor types cluster distinctly from electrogenic fishes with only ampullary electroreceptors (*Synodontis* siluriforms). The first principal component (PC1) explained 92.02% of the variation in brain region volumes and is strongly correlated with total brain volume ($\rho = -0.99$, p<10$^{-16}$). PC2 explained 4.81% of the total variation, which is 60.3% of the variation in region volumes not explained by total brain volume. Whereas all of the brain regions loaded in the same direction for PC1, cerebellum, torus semicircularis, and hindbrain loaded negatively on PC2 while the remaining regions (telencephalon, rest of brain, optic tectum, and olfactory bulbs) loaded positively. This suggests that concerted brain evolution explains the most variation in region volumes as seen in PC1, but that mosaic brain evolution could be contributing to the observed variation in brain region volumes as seen in PC2.

To ensure that differences in size ranges between regions were not biasing the results, we z-score-normalized the region volumes, reran the pPCA, and found nearly identical results (*Figure 4—figure supplement 1*). As there are multiple approaches to multivariate clustering analyses, we also ran a phylogenetic flexible discriminant analysis (pFDA) to investigate whether convergence in brain structure across electrosensory phenotypes persists irrespective of method. Again, we find distinct clustering between electrogenic species with both electroreceptor types, electrogenic species with only ampullary electroreceptors, and non-electric species, indicating three distinct electrosensory-associated

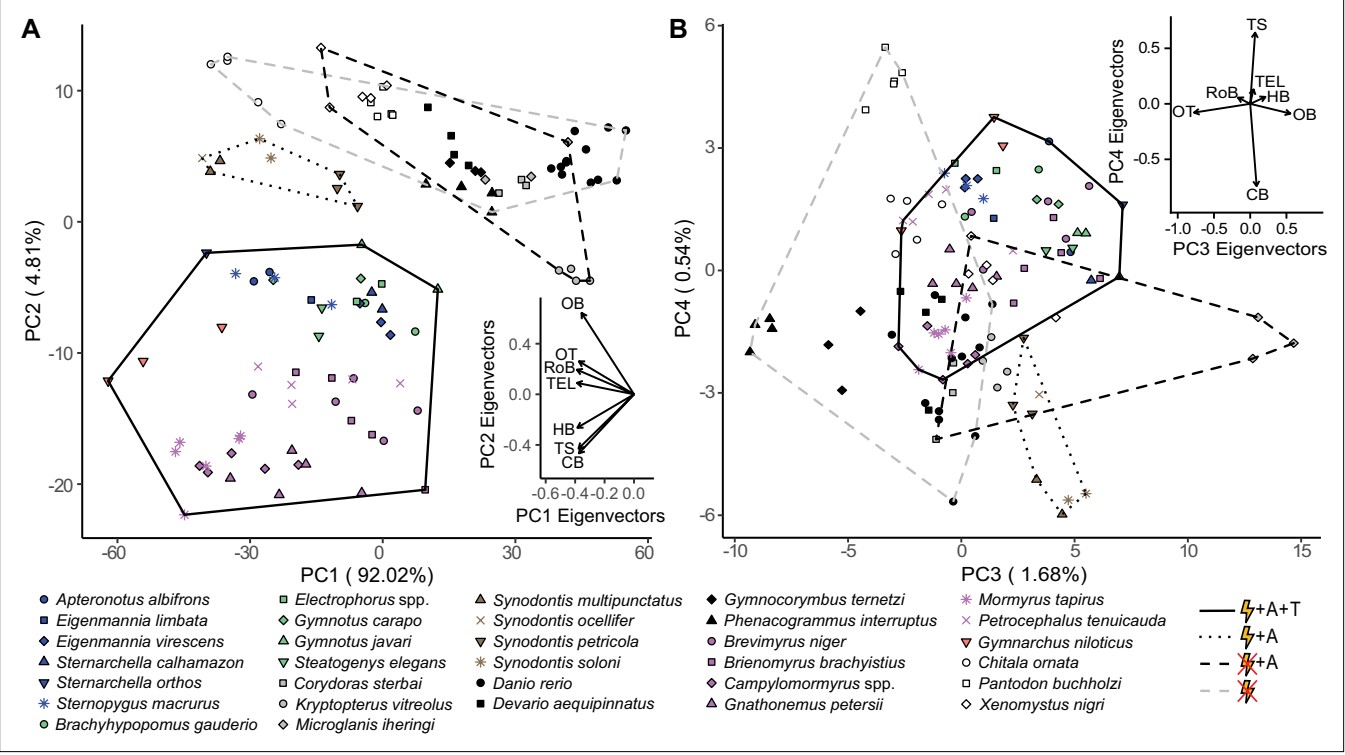

**Figure 4.** Species cluster distinctly in principal component (PC) space based on electrosensory phenotype. Hindbrain, torus semicircularis, and cerebellum are loaded in the direction of electrogenic taxa for PC1 and PC2 (**A**), but not for PC3 and PC4 (**B**). Each point represents an individual, shapes correspond to species, and colors correspond to lineages: orange, wave mormyroid (N = 1); pink, pulse mormyroids (N = 6); white, outgroup osteoglossiforms (N = 3); blue, wave gymnotiforms (N = 6); green, pulse gymnotiforms (N = 5); brown, *Synodontis* siluriforms (N = 4); gray, non-electric siluriforms (N = 3); black, outgroup otophysans (N = 4). Minimum convex hulls correspond to electrosensory phenotypes: electrogenic + ampullary + tuberous electroreceptors (solid), electrogenic + only ampullary electroreceptors (dotted), only ampullary electroreceptors (black dashed), and non-electrosensory (gray dashed). Insets shows PC eigenvectors of each brain region. OB, olfactory bulbs; TEL, telencephalon; HB, hindbrain; OT, optic tectum; TS, torus semicircularis; CB, cerebellum; RoB, rest of brain.

The online version of this article includes the following source data and figure supplement(s) for figure 4:

**Source data 1.** Phylogenetically corrected principal components analysis (pPCA) loadings for each non-normalized brain region.

**Source data 2.** Phylogenetically corrected principal components analysis (pPCA) phylogenetic generalized least squares (PGLS) model selection results for non-normalized data (N = 31).

**Source data 3.** Phylogenetic flexible discriminant analysis (pFDA) results table showing the regional coefficients for each discriminant axis and the means for each electrosensory phenotype group along each axis: electrogenic + tuberous and ampullary electroreceptors (E + A + T), electrogenic + only ampullary electroreceptors (E + A), and non-electric (Not E).

**Figure supplement 1.** Species cluster distinctly in z-score normalized principal component (PC) space based on electrosensory phenotype.

**Figure supplement 2.** Species cluster distinctly in discriminant space based on electrosensory phenotype.

cerebrotypes (*Clark et al., 2001*). The resulting three discriminant functions (i.e., number of electrosensory phenotype groups – 1) of the observed data accurately predicted electrosensory phenotype from the residuals of brain characters for all 32 species, further suggesting a relationship between electrosensory phenotypes and the observed brain region volume variation (*Figure 4—figure supplement 2*, *Figure 4—source data 3*). As all three clustering approaches demonstrated the same conclusions, we proceeded with the non-normalized pPCA.

To assess the relative importance of electrosensory phenotypes in explaining the axes of brain structural variation (PCs 1–4), we ran candidate models that considered body mass, total brain volume, presence or absence of electrogenesis, and electroreceptor type (tuberous and ampullary vs. only ampullary vs. none). Models that only consider allometric scaling with body mass and total brain volume would be consistent with concerted evolution while models that also consider either one or both electrosensory phenotypes would be in line with mosaic brain evolution since more than

just allometric scaling explains the observed variation in brain region volumes. Since PC1 strongly correlates with brain size, we removed total brain volume as a variable from all PC1 models.

We found that the model that considers the electrogenesis phenotype better explained PC1, but is statistically indistinguishable from the concerted model (*Figure 4—source data 2*), which further supports the role of concerted evolution in determining the sizes of individual brain regions. The model that considers electroreceptor type better explained PC2 (*Figure 4—source data 2*), which supports our hypothesis that the electrosensory system is related to mosaic evolutionary changes in brain region scaling. PC3 explains 1.68% of the total variation (21.1% of the variation not explained by total brain volume) and largely reflects the variation between olfactory bulbs and optic tectum with no separation between electrosensory phenotypes (*Figure 4B*). The model that considers electroreceptor type better predicts PC3 but is statistically indistinguishable from the concerted model, suggesting that this axis of brain variation likely evolved concertedly with brain size (*Figure 4—source data 2*). PC4 explains 0.54% of the total variation (6.8% of the variation not explained by total brain volume), largely reflects the variation between torus semicircularis and cerebellum, and is better predicted by the model that considers both electrosensory phenotypes (*Figure 4B*, *Figure 4—source data 2*). Taken together, these results highlight the importance of both concerted and mosaic brain evolution in producing the observed variation in brain region volumes.

## Mosaic shifts in electrogenic species relative to non-electric species

To directly test for mosaic shifts associated with electrosensory phenotypes, we fit PGLS regressions for each brain region against total brain size for species with each electrosensory phenotype and used an ANCOVA to test for significant differences in both slope and y-intercept of the PGLS relationships for each brain region. Considering the debate on how best to assess patterns of brain region scaling (*Finlay and Darlington, 1995*; *Yopak et al., 2010*), we also determined PGLS relationships for each brain region against total brain volume minus the focal brain region (remaining brain volume).

Since electroreceptive-only species and non-electrosensory species overlapped considerably in the pPCA, we combined all non-electric taxa and performed a phylogenetically corrected ANCOVA with phylogenetically corrected pairwise post-hoc testing to compare electrogenic taxa with both tuberous and ampullary electroreceptors (mormyroids and gymnotiforms), electrogenic taxa with only ampullary electroreceptors (*Synodontis* siluriforms), and non-electric taxa (*Figure 5*, *Figure 5—figure supplement 1*, *Figure 5—source data 1*, *Figure 5—source data 2*, *Figure 5—source data 3*, *Figure 5—source data 4*, *Figure 5—source data 5*, *Figure 5—source data 6*). We found a significant increase in y-intercept in cerebellum for all electrogenic species relative to non-electric species ($p<0.05$). For torus semicircularis, we found a significant increase in y-intercept in electrogenic + ampullary-only taxa relative to non-electric taxa ($p<0.05$) and a further increase in y-intercept in electrogenic + ampullary + tuberous taxa ($p<10^{-4}$). We found a significant increase in y-intercept in hindbrain for electrogenic + ampullary + tuberous taxa relative to non-electric taxa ($p<0.01$) with electrogenic + ampullary-only taxa being intermediate ($p>0.05$). These results were the same for both regressions against total brain volume (*Figure 5*, *Figure 5—source data 1*, *Figure 5—source data 2*) and regressions against remaining brain volume (*Figure 5—figure supplement 1*, *Figure 5—source data 4*, *Figure 5—source data 5*, *Figure 5—source data 6*).

There were significant decreases in y-intercept in olfactory bulbs and rest of brain for electrogenic + ampullary + tuberous taxa relative to both electrogenic + ampullary-only and non-electric taxa when regressed against total brain volume ($p_{OB}<0.05$, $p_{RoB}<0.05$). When regressed against remaining brain volume, we found similar results for olfactory bulbs ($p<0.05$), but we only found a significant decrease in y-intercept for electrogenic + ampullary + tuberous taxa relative to non-electric taxa in rest of brain ($p<10^{-4}$). For optic tectum, we found a significant decrease in y-intercept in electrogenic + ampullary + tuberous taxa relative to non-electric taxa ($p<10^{-4}$) with electrogenic + ampullary-only taxa as intermediate ($p>0.05$) when regressed against total brain volume. When regressed against remaining brain volume, we found a significant decrease in optic tectum y-intercept for electrogenic + ampullary + tuberous taxa relative to both electrogenic + ampullary-only and non-electric taxa ($p<0.05$). There were no significant differences in y-intercept for telencephalon when regressed against either total brain volume or remaining brain volume ($p>0.05$).

These results show that there are similar mosaic shifts in lineages that independently evolved electrogenesis regardless of analysis method. We also found a significant increase in slope in cerebellum

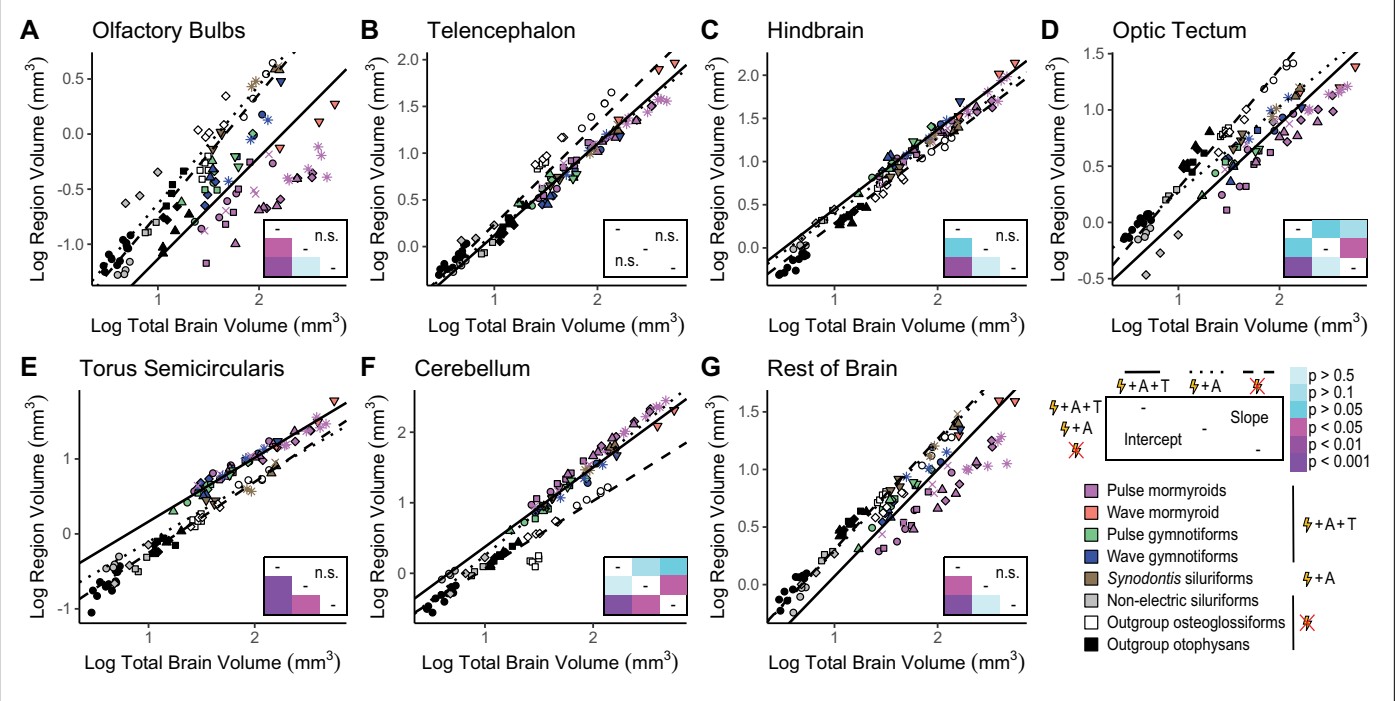

**Figure 5.** Mosaic increases in hindbrain, torus semicircularis, and cerebellum in electrogenic species with ampullary and tuberous electroreceptors. Plots of log region volume against log total brain volume for olfactory bulbs (**A**), telencephalon (**B**), hindbrain (**C**), optic tectum (**D**), torus semicircularis (**E**), cerebellum (**F**), and rest of brain (**G**). Each point corresponds to an individual and shapes represent the same species as *Figure 4*. Phylogenetic generalized least squares (PGLS) lines were determined from species means and correspond to electrosensory phenotypes that cluster distinctly in *Figure 4*: electrogenic + ampullary + tuberous electroreceptors (pink, orange, green, blue points; solid line; N = 18), electrogenic + only ampullary electroreceptors (brown points; dotted line; N = 4), and non-electric (gray, white, black points; dashed line; N = 10). Inset shows a heatmap of the phylogenetically corrected pairwise post-hoc analysis of covariance (ANCOVA) results with a Bonferroni correction for differences in intercept below the diagonal and differences in slope above the diagonal for each brain region. Significant differences are in shades of magenta/purple, and nonsignificant differences are in shades of blue. Post-hoc tests were not performed when the ANCOVA revealed no significant differences (indicated by 'n.s.').

The online version of this article includes the following source data and figure supplement(s) for figure 5:

**Source data 1.** Results of an analysis of covariance (ANCOVA) comparing phylogenetic generalized least squares (PGLS) relationships of region volume against total brain volume for each electrosensory phenotype: slope p, intercept p, and Pagel's lambda ($\lambda$).

**Source data 2.** Results (p-values) of pairwise post-hoc tests with a Bonferroni correction for an analysis of covariance (ANCOVA) comparing phylogenetic generalized least squares (PGLS) relationships of region volume against total brain volume for each electrosensory phenotype: electrogenic + tuberous and ampullary electroreceptors (E + A + T), electrogenic + only ampullary electroreceptors (E + A), and non-electric (Not E).

**Source data 3.** Estimated effect sizes (Cohen's d) of each contrast for an analysis of covariance (ANCOVA) comparing phylogenetic generalized least squares (PGLS) relationships of region volume against total brain volume for each electrosensory phenotype: electrogenic + tuberous and ampullary electroreceptors (E + A + T), electrogenic + only ampullary electroreceptors (E + A), and non-electric (Not E).

**Source data 4.** Results of an analysis of covariance (ANCOVA) comparing phylogenetic generalized least squares (PGLS) relationships of region volume against total brain–region volume for each electrosensory phenotype: slope p, intercept p, and Pagel's lambda ($\lambda$).

**Source data 5.** Results (p-values) of pairwise post-hoc tests with a Bonferroni correction for an analysis of covariance (ANCOVA) comparing phylogenetic generalized least squares (PGLS) relationships of region volume against total brain–region volume for each electrosensory phenotype: electrogenic + tuberous and ampullary electroreceptors (E + A + T), electrogenic + only ampullary electroreceptors (E + A), and non-electric (Not E).

**Source data 6.** Estimated effect sizes (Cohen's d) of each contrast for an analysis of covariance (ANCOVA) comparing phylogenetic generalized least squares (PGLS) relationships of region volume against total brain–region volume for each electrosensory phenotype: electrogenic + tuberous and ampullary electroreceptors (E + A + T), electrogenic + only ampullary electroreceptors (E + A), and non-electric (Not E).

**Figure supplement 1.** Apparent mosaic shifts in olfactory bulbs, hindbrain, optic tectum, torus semicircularis, cerebellum, and rest of brain between electrogenic species with ampullary and tuberous electroreceptors and non-electric species.

**Figure supplement 2.** No evidence of mosaic shifts in cerebellum or medulla of electrogenic chondrichthyans.

and a significant decrease in slope in optic tectum for electrogenic + ampullary-only species relative to non-electric species when regressed against total brain volume ($p_{CB}<0.05$, $p_{OT}<0.05$), which may be related to the reduced species sampling and brain size distribution of electrogenic + ampullary-only species (N = 4) relative to non-electric (N=10) and electrogenic + ampullary + tuberous species (N = 18). When regressed against remaining brain volume, we found a significant decrease in slope in cerebellum for non-electric taxa relative to both electrogenic + ampullary + tuberous taxa and electrogenic + ampullary-only taxa, which may be related to the obvious non-electric outlier *Pantodon buchholzi*. We did not find a significant difference in optic tectum slope when regressed against remaining brain volume ($p>0.1$). All other slope comparisons were nonsignificant for both analyses ($p>0.1$).

To assess how different brain regions covary with respect to electrosensory phenotype, we fit PGLS regressions for each region-by-region comparison and performed phylogenetically corrected ANCOVAs with phylogenetically corrected pairwise post-hoc testing for the same electrosensory phenotype groups (*Figure 6*). For electrogenic + ampullary + tuberous taxa relative to non-electric taxa, we found significant differences in y-intercept for olfactory bulbs against telencephalon ($p<0.05$); rest of brain against cerebellum ($p<10^{-3}$); telencephalon against olfactory bulbs ($p<0.01$), hindbrain ($p<0.01$), and cerebellum ($p<0.01$); hindbrain against optic tectum ($p<10^{-4}$) and telencephalon ($p<10^{-3}$); torus against hindbrain ($p<0.01$); and cerebellum against olfactory bulbs ($p<10^{-3}$), optic tectum ($p<10^{-4}$), and rest of brain ($p<10^{-4}$). For electrogenic + ampullary + tuberous taxa relative to both electrogenic + ampullary-only and non-electric taxa, we found significant differences in y-intercept for olfactory bulbs against hindbrain ($p<0.01$), torus ($p<0.01$), and cerebellum ($p<0.05$); optic tectum against hindbrain ($p<0.01$), torus ($p<10^{-3}$), and cerebellum ($p<0.05$); rest of brain against hindbrain ($p<0.01$) and torus ($p<10^{-4}$); telencephalon against torus ($p<0.01$); hindbrain against olfactory bulbs ($p<0.05$) and rest of brain ($p<0.01$); and torus against olfactory bulbs ($p<0.01$), optic tectum ($p<10^{-3}$), rest of brain ($p<10^{-4}$), and telencephalon ($p<0.05$). For both electrogenic + ampullary + tuberous taxa and electrogenic + ampullary-only relative to non-electric taxa, we found significant differences in y-intercept for cerebellum against telencephalon ($p<0.05$) and in slope for rest of brain against optic tectum ($p<0.05$); telencephalon against optic tectum ($p<0.05$); hindbrain against optic tectum ($p<0.01$); torus against optic tectum ($p<0.01$); and cerebellum against optic tectum ($p<0.01$) and telencephalon ($p<0.05$). All remaining comparisons for y-intercept and slope were nonsignificant ($p>0.05$).

Taken together, these results suggest that there are two covarying brain structure groupings, with olfactory bulbs, optic tectum, telencephalon, and rest of brain falling into one group and hindbrain, torus, and cerebellum falling into the other group as all comparisons of group 1 regions against group 2 regions have significant mosaic shifts between electrosensory phenotypes. However, this pattern is strongest for tuberous receptor taxa relative to non-electric taxa. These findings mirror the pPCA results, in which these taxa were separated primarily by PC2, for which olfactory bulbs, optic tectum, telencephalon, and rest of brain loaded positively, whereas hindbrain, torus, and cerebellum all loaded negatively (*Figure 4*). Differences in slope were only found in regions where there are obvious outliers in the non-electric fishes (cerebellum: *P. buchholzi*; optic tectum: *Microglanis iheringi*), suggesting that these species might have mosaic shifts relative to other non-electric osteoglossiforms and otophysans, respectively, but additional studies are needed to investigate this possibility further. We did not find any slope differences in the olfactory bulb comparisons, which also have an obvious outlier (*M. iheringi*); however, there is more variability in olfactory bulbs compared to the other brain regions, which may decrease the influence of outliers.

## Lineage-specific mosaic shifts within electrosensory phenotypes

To determine if there are lineage-specific differences within electrosensory phenotypes, we performed a phylogenetically corrected ANCOVA of each brain region between mormyroids and gymnotiforms (*Figure 7A*, *Figure 7—figure supplement 1A*, *Figure 7—source data 1*, *Figure 7—source data 2*) and between non-electric osteoglossiforms and otophysans (*Figure 7*, *Figure 7—figure supplement 1B*, *Figure 7—source data 1*, *Figure 7—source data 2*). When regressed against total brain volume, we found a shallower slope and smaller y-intercept for mormyroids in olfactory bulbs ($p_{slope}<10^{-4}$, $p_{intercept}<0.05$) and rest of brain ($p_{slope}<10^{-2}$, $p_{intercept}<0.05$), shallower slope for mormyroids in torus semicircularis ($p<10^{-2}$), and steeper slope for mormyroids in cerebellum ($p<0.05$). We found similar results when regressed against remaining brain volume, but we did not find a significant difference in cerebellum slope between mormyroids and gymnotiforms ($p>0.05$). All remaining comparisons

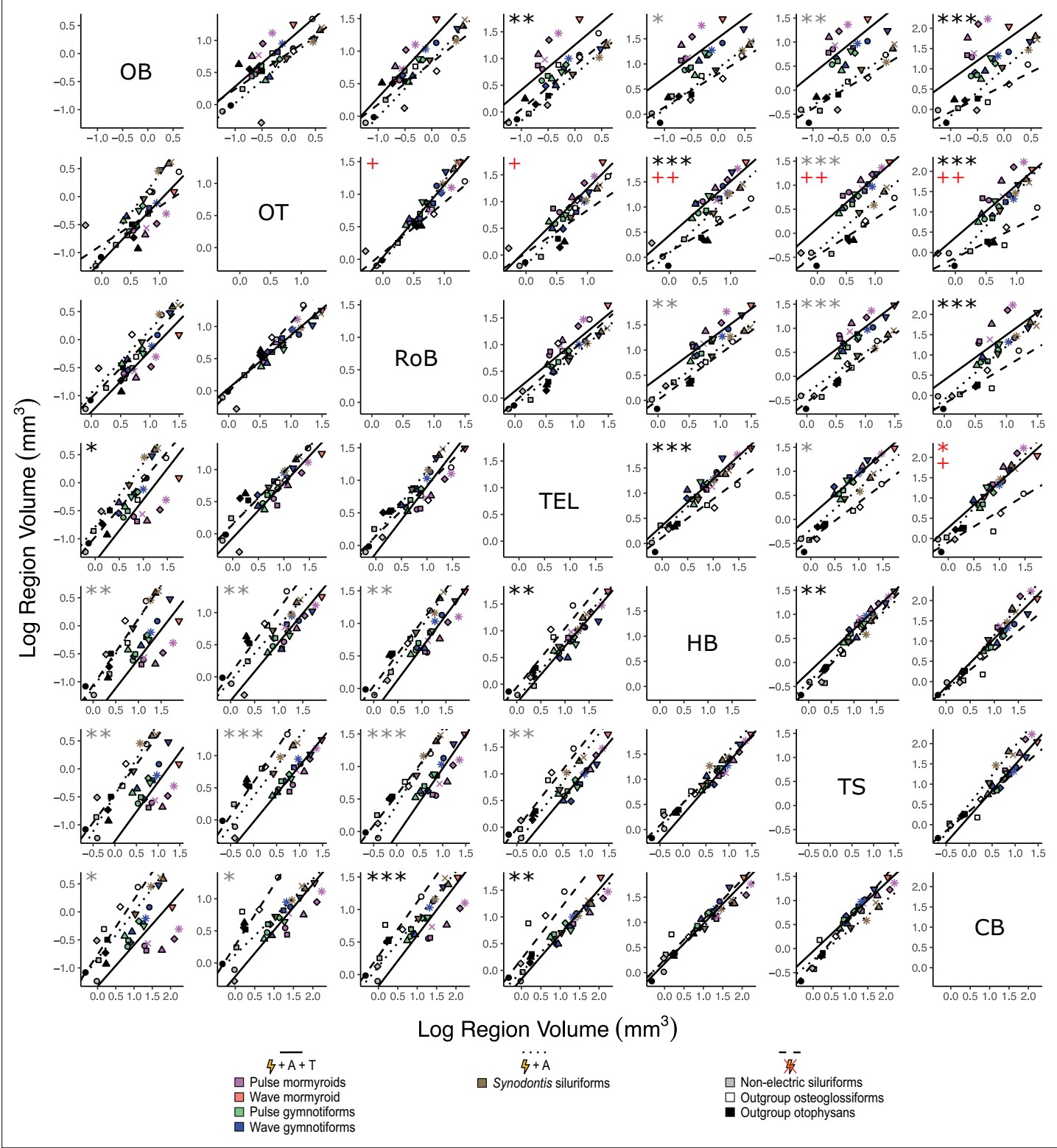

**Figure 6.** Matrix of scatterplots for each region-by-region comparison. Columns indicate the region on the y-axis, and rows indicate the region on the x-axis. Points correspond to species means, and shapes represent the same species as *Figure 4*. Phylogenetic generalized least squares (PGLS) lines were determined from species means and correspond to electrosensory phenotypes that cluster distinctly in *Figure 4*: electrogenic + tuberous and ampullary electroreceptors (pink, orange, green, blue points; solid line; N = 18), electrogenic + only ampullary electroreceptors (brown points; dotted line; N = 4), and non-electric (gray, white, black points; dashed line; N = 10). Significant differences in intercept are marked with asterisks: *p<0.05, **p<0.01.

*Figure 6 continued on next page*

*Figure 6 continued*

Significant differences in slope are marked with pluses: +p<0.05, ++p<0.01, +++p<0.001. Significant differences between electrogenic + tuberous and ampullary electroreceptors taxa and non-electric taxa are marked in black. Significant differences between tuberous receptor taxa (E + A + T) and taxa lacking tuberous receptors (E + A and not E) are marked in gray. Significant differences between all electrogenic taxa (E + A + T and E + A) and non-electric taxa are marked in red. OB, olfactory bulbs; TEL, telencephalon; HB, hindbrain; OT, optic tectum; TS, torus semicircularis; CB, cerebellum; RoB, rest of brain.

The online version of this article includes the following source data for figure 6:

**Source data 1.** Matrix of region-by-region analysis of covariance (ANCOVA) results comparing phylogenetic generalized least squares (PGLS) relationships for each electrosensory phenotype: slope p (S), intercept p (I), and Pagel's lambda (L).

**Source data 2.** Matrix of region-by-region analysis of covariance (ANCOVA) results comparing phylogenetic generalized least squares (PGLS) relationships for each electrosensory phenotype: electrogenic + tuberous and ampullary electroreceptors (E + A + T), electrogenic + only ampullary electroreceptors (E + A), and non-electric (Not E).

**Source data 3.** Matrix of region-by-region analysis of covariance (ANCOVA) results comparing phylogenetic generalized least squares (PGLS) relationships for each electrosensory phenotype: electrogenic + tuberous and ampullary electroreceptors (E + A + T), electrogenic + only ampullary electroreceptors (E + A), and non-electric (Not E).

for y-intercept and slope were nonsignificant for both analyses (p>0.05). For the region-by-region comparisons, we did not find the same groupings as across electrosensory phenotypes (*Figure 7—figure supplement 2*, *Figure 7—source data 3*), further suggesting that the observed differences in brain structure are related to evolving electrosensory systems. We did find significant differences in y-intercept and/or slope in most olfactory bulb comparisons (p<0.05), except olfactory bulbs and torus comparisons and optic tectum against olfactory bulbs (p>0.05). We also found significant differences in y-intercept for the telencephalon and torus versus rest of brain comparisons (p<0.05) and in slope for some comparisons between optic tectum, hindbrain, torus, and cerebellum (p<0.05). These differences are likely contributing to the secondary clustering between mormyroids and gymnotiforms in the pPCA (*Figure 4A*) and suggest that there are more nuanced distinctions in brain structure between these two lineages. For non-electric fishes, we found that osteoglossiforms have a significantly larger y-intercept for telencephalon ($p_{TEL}<10^{-2}$), but no differences in either slope or y-intercept in the other brain regions (p>0.1) when regressed against both total brain volume (*Figure 7B*, *Figure 7—source data 1*) and remaining brain volume (*Figure 7—figure supplement 1B*, *Figure 7—source data 2*). We found that osteoglossiforms also have significantly larger y-intercepts for telencephalon against all other regions (p<0.05) and for torus against cerebellum (p<0.05), but no differences in either slope or y-intercept in the other comparisons (p>0.05; *Figure 7—figure supplement 3*, *Figure 7—source data 4*).

Some gymnotiforms produce continuous electrical discharges with variable frequency (wave-type), while others produce discrete electric discharges separated by variable periods of stasis (pulse-type). To assess whether there are mosaic shifts associated with the evolution of electrical discharge type, we performed a phylogenetically corrected ANCOVA between wave-type and pulse-type gymnotiforms (*Figure 7C*, *Figure 7—figure supplement 1C*, *Figure 7—source data 1*, *Figure 7—source data 2*). We found significant increases in slope for wave gymnotiforms in olfactory bulbs against hindbrain and cerebellum and in telencephalon against hindbrain and cerebellum (p<0.05; *Figure 7—figure supplement 4*, *Figure 7—source data 5*). However, we found no differences in either y-intercept or slope of any region against total brain volume between wave and pulse gymnotiforms and only a significant increase in slope in telencephalon against remaining brain volume for wave gymnotiforms (p<0.05), suggesting that the transition between discharge types did not relate to brain structure at this scale. It was not possible to directly test this in mormyroids since there is only one wave-type mormyroid species (*Gymnarchus niloticus*) that is sister to the family of all pulse-type mormyroids. However, the electrosensory system of the wave-type mormyroid is similar to that of gymnotiforms (*Bell and Maler, 2005*), and the wave mormyroid tends to be more similar to gymnotiforms in both brain region residuals (*Figure 7A*) and the pPCA (*Figure 4*). After excluding the wave mormyroid, we found the same regional differences associated with all mormyroids, along with a significant increase in the cerebellum y-intercept in pulse mormyroids relative to gymnotiforms (p<0.05), suggesting that the extraordinarily enlarged cerebellum of some mormyroid species is the result of both a steeper allometric relationship and a mosaic shift in the ancestor of pulse mormyroids.

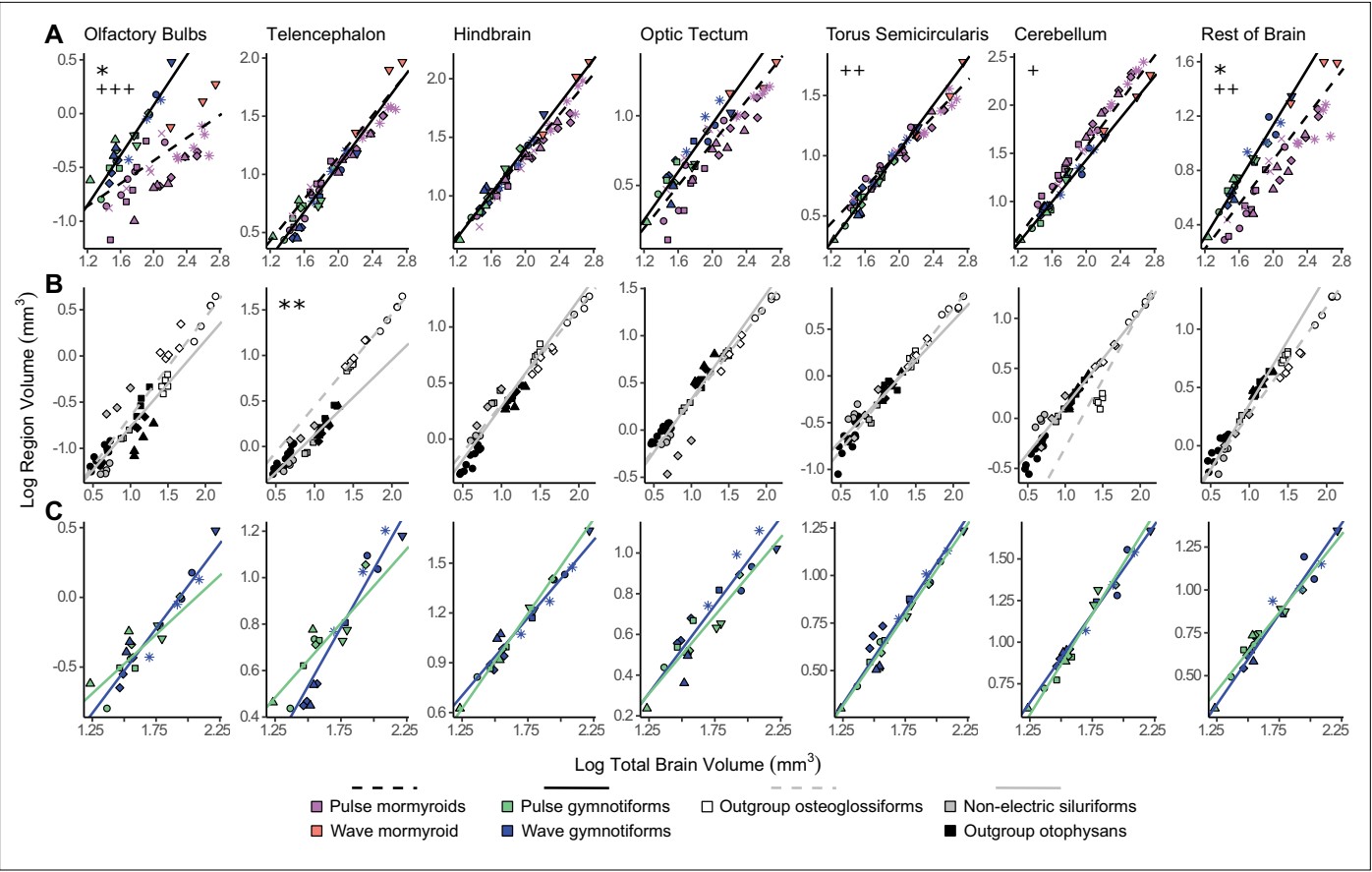

**Figure 7.** Lineage-specific mosaic shifts within phenotypes in olfactory bulbs, telencephalon, and rest of brain. Plots of log brain region volumes by log total brain volume for (**A**) mormyroids (pink and orange points, dashed black lines, N = 7) vs. gymnotiforms (blue and green points, solid black line, N = 11); (**B**) non-electric osteoglossiforms (white points, dashed gray line, N = 3) vs. non-electric otophysans (gray and black points, solid gray line, N = 7); and (**C**) wave (blue, N = 6) vs. pulse (green, N = 5) gymnotiforms. Each point corresponds to an individual, and shapes represent the same species as *Figure 4*. Phylogenetic generalized least squares (PGLS) lines were determined from species means and compared using phylogenetically corrected analysis of covariances (ANCOVAs). Significant differences in intercept are marked with asterisks: *p<0.05, **p<0.01. Significant differences in slope are marked with pluses: +p<0.05, ++p<0.01, +++p<0.001.

The online version of this article includes the following source data and figure supplement(s) for figure 7:

**Source data 1.** Lineage analysis of covariance (ANCOVA) results comparing phylogenetic generalized least squares (PGLS) relationships of region volume against total brain volume for each lineage: slope p, intercept p, Pagel's lambda ( λ ), Cohen's d.

**Source data 2.** Lineage analysis of covariance (ANCOVA) results comparing phylogenetic generalized least squares (PGLS) relationships of region volume against total brain–region volume for each lineage: slope p, intercept p, Pagel's lambda ( λ ), Cohen's d.

**Source data 3.** Matrix of region-by-region analysis of covariance (ANCOVA) results comparing phylogenetic generalized least squares (PGLS) relationships for mormyroids vs. gymnotiforms: slope p (S), intercept p (I), Pagel's lambda (L), and Cohen's d (D).

**Source data 4.** Matrix of region-by-region analysis of covariance (ANCOVA) results comparing phylogenetic generalized least squares (PGLS) relationships for non-electric osteoglossiforms vs. non-electric otophysans: slope p (S), intercept p (I), Pagel's lambda (L), and Cohen's d (D).

**Source data 5.** Matrix of region-by-region analysis of covariance (ANCOVA) results comparing phylogenetic generalized least squares (PGLS) relationships for wave vs. pulse gymnotiforms: slope p (S), intercept p (I), Pagel's lambda (L), and Cohen's d (D).

**Figure supplement 1.** Lineage-specific mosaic shifts within phenotypes in olfactory bulbs, telencephalon, and rest of brain.

**Figure supplement 2.** Matrix of scatterplots for each region-by-region comparison for mormyroids (pink and orange points, dashed black lines, N = 7) vs. gymnotiforms (blue and green points, solid black line, N = 11).

**Figure supplement 3.** Matrix of scatterplots for each region-by-region comparison for non-electric osteoglossiforms (white points, dashed gray line, N = 3) vs. non-electric otophysans (gray and black points, solid gray line, N = 7).

**Figure supplement 4.** Matrix of scatterplots for each region-by-region comparison for wave (blue, N = 6) vs. pulse (green, N = 5) gymnotiforms.

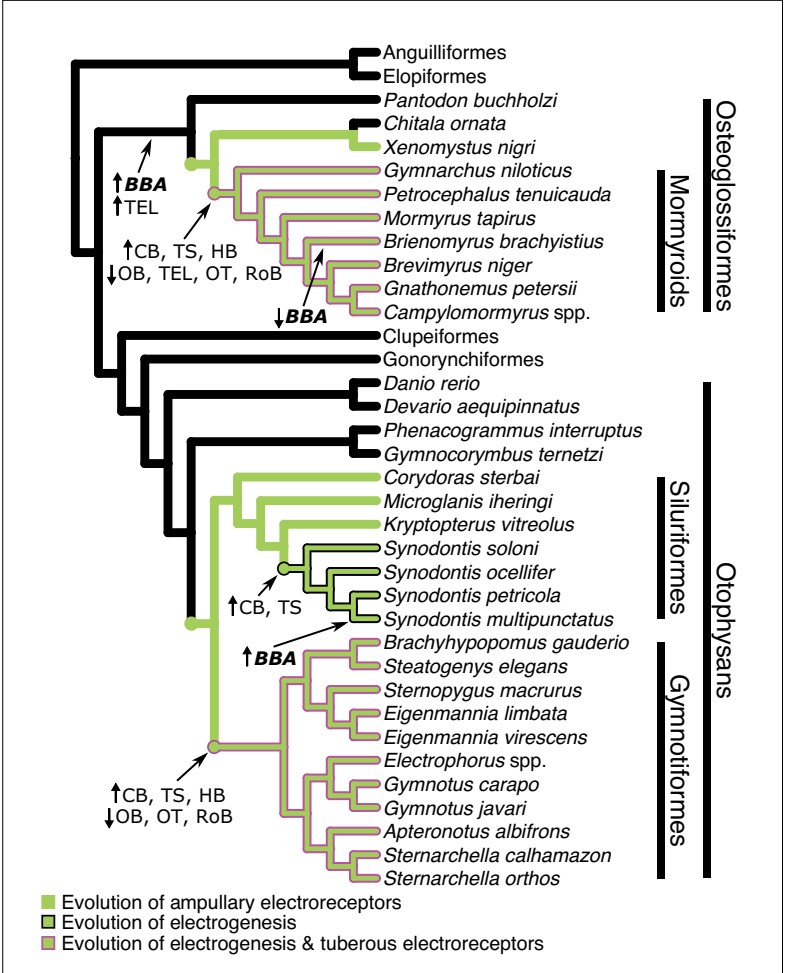

**Figure 8.** Cladogram of the inferred phylogenetic relationships of species included in this study (N = 32) and the orders between them depicting where shifts in brain–body allometries (indicated in bold italics) and relative mosaic shifts in region–brain allometries likely occurred. Order level relationships are based on *Hughes et al., 2018*. Green branches represent presence of ampullary electroreceptors. Black outline represents electrogenic species while the magenta outline represents electrogenic species with tuberous electroreceptors. BBA, brain body allometry; OB, olfactory bulbs; TEL, telencephalon; HB, hindbrain; OT, optic tectum; TS, torus semicircularis; CB, cerebellum; RoB, rest of brain.

## Discussion

We used osteoglossiform and otophysan fishes to test whether mosaic shifts in brain region volumes are associated with the convergent evolution of behaviorally novel active electrosensory systems. Although the mosaic shifts previously found in mormyroids were hypothesized to be related to the evolution of electrogenesis, it remained unknown if these patterns would be found in other electrogenic lineages. The brain scaling patterns of electrogenic versus non-electric osteoglossiforms and otophysans are strikingly similar despite the considerable phylogenetic distance between them, revealing distinct electrosensory associated cerebrotypes (*Figure 8*). Further, these electrosensory cerebrotypes converged independent of the variable brain–body allometric relationships of osteoglossiforms and otophysans (*Figure 3*).

Gymnotiforms have the most similar electrosensory system to mormyroids in terms of electrosensory structures, neural processing, and behavioral usage (*Hopkins, 1995*). In both of these lineages, we found mosaic increases in cerebellum, hindbrain, and torus semicircularis compared to their non-electric relatives, which suggests that these evolutionary shifts in brain structure largely reflect their coevolution with electrogenesis and tuberous electroreceptor phenotypes. First-order electrosensory processing takes place in the electrosensory lateral line lobe (ELL) of the hindbrain, which projects to

the torus for further processing of electrocommunication and electrolocation signals (*Baker et al., 2013*; *Bell and Maler, 2005*; *Metzen and Chacron, 2021*). Electrosensory information projects from the torus both directly and indirectly to areas of the cerebellum with large, reciprocal connections between cerebellum, torus, and the ELL of both mormyroids and gymnotiforms. Multiple areas of the mormyroid cerebellum show responses to electrosensory stimuli (*Russell and Bell, 1978*). More generally, cerebellum is also known to be involved in predicting the sensory consequences of motor movements and subsequent error detection, nonmotor functions, and learning (*Hull, 2020*; *Popa and Ebner, 2019*; *Strick et al., 2009*). The overwhelming evidence of feedback circuits between initial electrosensory processing regions and cerebellum in both mormyroids and gymnotiforms suggests that the cerebellum may also be involved in processing electrosensory information in addition to electromotor control (*Bell and Maler, 2005*; *Paulin, 1993*). The hindbrain is also involved in generating electromotor output (*Caputi et al., 2005*), which suggests that both electrosensory processing and electromotor control are related to evolutionary changes in relative region sizes.

Electrogenic *Synodontis,* which only have ampullary electroreceptors, have significant mosaic increases in cerebellum and torus semicircularis relative to non-electric fishes. *Synodontis* electrical discharges are likely detectable by their ampullary electroreceptors (*Hagedorn et al., 1990*; *Zupanc and Bullock, 2005*) and involved in electrocommunication (*Albert and Crampton, 2006*; *Boyle et al., 2014*), which could relate to enlargement of these regions relative to electroreceptive but non-electric species. We found that *Synodontis* were intermediate between electrogenic fishes with tuberous receptors and non-electric fishes in torus and hindbrain, although the difference in hindbrain was not significant despite the hindbrain's role in generating electromotor output in synodontids (*Hagedorn et al., 1990*; *Kéver et al., 2020*). The hindbrain also contains the facial and vagal lobes, which are enlarged in siluriforms and cypriniforms (*Striedter, 2005*), potentially obscuring a relationship with electrogenesis, and further highlighting the complexity of gross-scale brain region evolution.

The addition of tuberous electroreceptors increases the range of detectable signals and total electrosensory input to the brain relative to only ampullary electroreceptors (*Crampton, 2019*), and more subregions of the torus and hindbrain are devoted to electrosensory processing in tuberous electroreceptor species (*Bell and Maler, 2005*). The enlarged torus in *Synodontis* may also relate to acoustic communication. Some *Synodontis* species produce swim bladder sounds (*Boyle et al., 2014*), and the torus is involved in auditory processing (*Fay and Edds-Walton, 2008*). This is also true for mormyroids, as they are known to have specialized hearing, and some species produce acoustic signals (*Ladich and Winkler, 2017*). However, gymnotiforms are not known to produce acoustic signals, and all otophysans possess accessory structures that improve hearing (*Ladich and Winkler, 2017*). Additionally, *M. iheringi* and several *Corydoras* species are also known to utilize acoustic communication (*Kaatz et al., 2010*), but we did not find an enlarged torus in these taxa.

Interestingly, all electrogenic fishes have a significant mosaic increase in cerebellum regardless of electroreceptor type. Relative brain region sizes of electroreceptive only species are largely consistent with non-electrosensory species, which suggests that the evolution of electrogenesis strongly relates to structural brain composition. However, two chondrichthyan lineages have independently evolved electrogenesis, Torpediniformes and Rajidae (*Bennett, 1971*), with no evidence of mosaic shifts in cerebellar size in these taxa (*Mull et al., 2020*; *Yopak et al., 2010*; *Figure 5—figure supplement 2*). It is possible this is because chondrichthyan cerebellums are already massively enlarged compared to their closest relatives, agnathans, which arguably lack a proper cerebellum (*Striedter and Northcutt, 2020*). Unfortunately, the brain structure of other independently evolved electrogenic lineages, such as *Malapterurus* catfishes, *Astroscopus* stargazers, and *Uranoscopus* stargazers, remain unknown. The addition of these lineages could better elucidate the relationship between the evolution of electrogenesis and structural brain composition, especially as stargazers are the only electrogenic fishes that lack electroreceptors of any type.

Alternatively, it is possible the enlarged cerebellum in *Synodontis* is unrelated to the evolution of electrogenesis. Like *Synodontis*, Rajidae produce sporadic discharges likely used for electrocommunication, while mormyroids and gymnotiforms produce near continuous discharges (*Bennett, 1971*; *Crampton, 2019*). Considering we find a further enlargement in the cerebellum of pulse mormyroids which produce electrical discharges at varying and complex timing intervals, it is possible that the specific usage and complexities of electrical discharges may relate to the degree of cerebellar enlargement. We do not find any differences in cerebellar volume of wave and pulse gymnotiforms,

but pulse gymnotiforms produce discharges at regular intervals like wave-type fishes and unlike the irregularly discharging pulse mormyroids (*Caputi et al., 2005*). Further research on the usage of sporadic electrical discharges and the related electrosensory pathways are needed to better elucidate the relationship between electrogenesis and cerebellar enlargement. In particular, further research on *Synodontis* species is greatly warranted as electrogenesis in these species appears evolutionarily labile (i.e., some species produce continuous discharges, some produce sporadic discharges, and one species produced no discharges under experimental conditions), yet only 13 of 200+ species have been investigated for electrogenesis, and very little is known about electrosensory processing in these species (*Baron et al., 1994*; *Baron, 2002*; *Boyle et al., 2014*; *Hagedorn et al., 1990*). We also found a shift in the brain–body allometry of *S. multipunctatus*; however, we are unable to determine whether this is associated with any specific electrosensory phenotypes without additional research into more synodontid species. Our results only further highlight the value of these species in understanding the relationship between electrogenesis and brain evolution.

Electrosensory information also projects to the telencephalon (*Bell and Maler, 2005*), but we do not find a mosaic increase in any electrosensory taxa relative to non-electric taxa. This is likely because the telencephalon is involved in higher-order sensory integration across many different sensory modalities (*Striedter and Northcutt, 2020*) while sensory systems mostly remain segregated in the lower-order processing of hindbrain and torus (*Meek and Nieuwenhuys, 1998*). Surprisingly, we find that non-electric osteoglossiforms have a mosaic increase in telencephalon relative to non-electric otophysans. The telencephalon of osteoglossiforms is highly differentiated, more so than other teleosts (*Meek and Nieuwenhuys, 1998*), which suggests that osteoglossiforms evolved enlarged telencephalons relative to Elopomorpha, followed by a relative decrease in mormyroids alongside the evolution of electrogenesis and mosaic increases in other brain regions. Further research is needed to determine why osteoglossiforms have enlarged telencephalons.

Optic tectum is also involved in sensorimotor integration, particularly with respect to the electrosensory, lateral line, and visual systems, in addition to being the primary target of visual input to the brain (*Meek and Nieuwenhuys, 1998*). Yet we find that tuberous receptor fishes have a mosaic decrease in optic tectum relative to non-electric fishes while electrogenic + ampullary-only fishes are intermediate. Additionally, tuberous receptor fishes have a mosaic decrease in olfactory bulbs relative to fishes lacking tuberous receptors, which together could indicate decreased reliance on visual and olfactory systems. Gymnotiforms are thought to have poor vision (*Takiyama et al., 2015*), and different mormyroid lineages specialize to varying degrees in visual versus electrocommunication systems (*Stevens et al., 2013*). We do find a mosaic decrease in rest of brain in electrogenic taxa with tuberous electroreceptors relative to fishes lacking tuberous receptors and a mosaic decrease in rest of brain in mormyroids relative to gymnotiforms that could reflect a trade-off in one or more of the subregions that comprise the rest of brain, but since we combined these subregions, we are unable to speculate about their evolution.

Here, we assume that increased brain region volume corresponds to increased neuron number, which increases processing power and reflects behavioral changes that natural selection can act upon in one lineage with respect to another. However, increases in absolute regional volume can result from increased neuron number, neuron size, glia number, or any combination thereof (*Herculano-Houzel, 2012*; *Marhounová et al., 2019*). Previous studies found that neuron number and size tend to scale with brain size, but this scaling varies across both lineages and brain regions, with some regions having more neurons than expected given total brain size (*Barton, 2012*; *Herculano-Houzel et al., 2014*; *Kverková, 2022*; *Marhounová et al., 2019*). These findings suggest that volume measures could over- or underrepresent neuron number, and future studies should investigate the neuronal composition of these regions to better investigate how absolute region volumes and processing capabilities have changed.

Changes in relative regional volumes can result from any of the aforementioned mechanisms in the focal region, but they can also result from changes in other regions that cause a shift in the relative proportion of any given region. In particular, we want to emphasize that all of our identified mosaic shifts are relative to total brain size and to other lineages. An increase in the size of one region necessitates a decrease in one or more of the other brain regions since all regional measurements are relative to total brain size. For example, an increase in the number of neurons in the cerebellum of mormyroids would lead to an increase in absolute cerebellar volume. Even if the neuron number, size, and glial

content of the telencephalon remained constant in all osteoglossiforms and thus no changes in absolute telencephalon volume occurred, mormyroid telencephalon size would have necessarily decreased relative to total brain size since total brain size has increased with the addition of more cerebellar neurons. Thus, the relative proportion of telencephalon may have decreased in mormyroids relative to other osteoglossiforms due solely to increases in the absolute sizes of cerebellum, hindbrain, and torus. However, it is quite difficult to distinguish this scenario from the possibility that absolute telencephalon size has also changed in some manner, especially when other factors such as body size differ as well. Further, the patterns of regional scaling in one lineage are relative to others, and may reflect differences in relative investment across lineages.

Due to the complex evolutionary histories of different brain regions, subregions, and total brain size, we are unable to make any claims regarding the evolution of absolute region sizes at this phylogenetic scale. Even commonly used 'reference' brain regions such as our 'rest of brain' have subregions like the lateral hypothalamic regions and preglomerular complex that are known to vary tremendously in size across teleosts, and the preglomerular complex is extensive in mormyroids in particular (*Wullimann, 2020*; *Wullimann and Northcutt, 1990*). Regardless of the specific differences in absolute region volumes, we still find overwhelming evidence of relative differences in the volume of individual regions that, although the mechanism is currently unknown, are likely still reflecting biologically meaningful differences in relative neural processing investment across these fishes.

Evidence of mosaic evolution at smaller subregional, nuclei, and circuit levels is readily available (*Carlson et al., 2011*; *Vélez et al., 2017*; *Gutiérrez-Ibáñez et al., 2014*; *Moore and DeVoogd, 2017*; *DeCasien and Higham, 2019*; *Krebs, 1990*), but rarely have mosaic shifts been observed at the level of major brain regions and even less so alongside convergence in behavioral evolution. Our findings support the hypothesis that mosaic brain evolution occurs more readily under substantial selective pressure that favors a greater expansion of a particular brain region than allometric scaling can accommodate without incurring a substantial energetic cost (*Striedter and Northcutt, 2020*). Although not necessarily exclusive to electrosensing or sensory systems in general, we suspect the evolution of this novel sensory system provided such a strong selective pressure. Indeed, our clearest example of gross-scale mosaic evolution occurs alongside evolutionary changes in both the sensory and motor system, and we suggest looking towards other instances of behavioral novelty for additional potential examples of gross-scale mosaic brain evolution.

However, the evolutionary pressures on brain structure are multifaceted. As tasks require further integration of different areas of the brain, selective pressure favoring one trait could instead lead to coordinated selection favoring the expansion of all regions (*Avin et al., 2021*; *Striedter and Northcutt, 2020*). Indeed, we find that multiple regions covary with electrosensory phenotypes (*Figure 6*). Not only are there evolutionary forces acting on brain region scaling to consider, but also the evolutionary forces acting upon total brain size. After constraint on total brain size was added in a barebones model of brain region evolutionary dynamics, the probability of mosaic evolution increased under most tested scenarios (*Avin et al., 2021*). The decoupling of brain–body allometries has been reported in birds and mammals, whereas actinopterygians are consistently found to have more constrained allometric relationships, whether through explicit constraints or strong stabilizing selection (*Tsuboi, 2021*; *Tsuboi et al., 2018*). Thus, it is entirely possible that the interactions of both region size and total brain size scaling may increase the likelihood of gross-scale mosaic brain evolution in fishes relative to birds and mammals. This may reflect differences in evolutionary strategies to enlarge brain size in conjunction with indeterminant and determinant growth, respectively. However, more research is needed on the evolution of brain–body and region–brain allometries in other lineages with indeterminant growth.

Our results also highlight the importance of considering differences in allometric slope and relaxing the assumption of shared allometric relationships for major taxonomic groups. With brain–body allometries, average slopes across major vertebrate taxonomic levels (class to genus) are relatively constant (*Tsuboi et al., 2018*); however, when not a priori defining grades based on strict taxonomic-level distinctions, significant differences in the allometric relationships of various groups at different taxonomic levels are readily detected (*Ksepka et al., 2020*; *Smaers et al., 2021*; *Figure 3*). Indeed, when we allow only intercept to vary between grades while assuming parallel slopes, we no longer detect any reliable grade shifts within actinopterygians. Even when allowing slope to vary, two of our

identified shifts were undetectable when assuming the brain–body allometry is constant within orders despite analyzing much of the same data (*Tsuboi, 2021*).

Like previous work in other lineages (*Finlay and Darlington, 1995*; *Yopak et al., 2010*), we also find differences in slope across region–brain allometries at varying taxonomic levels, indicating that the dramatic differences in observed region volumes can result from both mosaic shifts between lineages and evolutionary changes in total brain size within a lineage (*Figure 5—source data 1*, *Figure 7—source data 1*). As interspecific (evolutionary) allometries are an emergent property of developmental (within individuals) and static (within species) allometries (*Pélabon et al., 2014*; *Tsuboi, 2021*), it is difficult to assess the mechanism leading to these differences in slope across species without further research into evolutionary changes in developmental and static allometries. Previous work considering the effect of static allometries on evolutionary allometries found that evolutionary changes in both the static slope and intercept are contributing to the steeper evolutionary slope found across osteoglossiforms (*Tsuboi, 2021*). Steeper static slopes may indicate a higher rate of brain growth to body growth in adult stages while larger static intercepts might reflect increased brain mass at the transition between embryonic and juvenile growth phases in fishes (*Oikawa et al., 1992*; *Oikawa and Itazawa, 1984*; *Tsuboi, 2021*). Previous work in marsupials did not find any mechanistic links between regional neurogenesis timing or growth rate and static or evolutionary region–brain allometric differences in either slope or intercept despite finding extensive heterochronic differences between species (*Carlisle et al., 2017*). This finding suggests that the intraspecific mechanisms resulting in these interspecific scaling differences may differ across species, but additional research is needed to determine whether there are shared intraspecific mechanisms resulting in interspecific differences in slope versus intercept in other lineages, especially in those with extensive adult neurogenesis.

Different lineages can independently evolve the same phenotype via the same mechanism (parallel evolution) or different mechanisms (convergent evolution). Given the phylogenetic distance between osteoglossiforms and otophysans, it would be more remarkable to find that the different electrosensory systems and mosaic shifts in brain region volumes evolved in parallel rather than by convergence. Given that the mechanism of these regional increases remains unknown, we argue for a more conservative assumption of convergent evolution for electrosensory cerebrotypes. This is supported by the fact that the cerebellar subregion that has expanded the most in mormyroids is the valvula cerebelli while in gymnotiforms, it is the corpus cerebelli (*Meek and Nieuwenhuys, 1998*). Additionally, the torus is laminar in gymnotiforms while there are distinct nuclei in the non-laminar torus of mormyroids (*Bell and Maler, 2005*), and we find evidence of a steeper brain–body allometric relationship for osteoglossiforms that appears unrelated to the evolution of an electrosensory system in mormyroids. Interestingly, we found a subsequent decrease in the brain–body allometry of two small-brained sister mormyroids (*I. henryi* and *B. brachyistius*), which further suggests the steeper brain–body allometry of osteoglossiforms is unrelated to electrosensory capabilities. Further research is needed into osteoglossiforms as a whole to determine why this lineage has an increased brain–body allometry. Regardless of whether the mechanism of relative regional scaling evolution is convergent or parallel, we provide evidence of repeated, independent mosaic evolution of major brain regions in association with a convergent behavioral novelty. These findings demonstrate that evolutionary changes in gross-scale brain structure are surprisingly predictable alongside the evolution of active electrosensory systems, even when the underlying brain–body allometry differs. More broadly, these findings suggest that mosaic brain evolution may occur alongside the evolution of behavioral novelty and could reflect a degree of predictability in brain evolution with behavioral evolution, especially when constraint in brain–body allometries are strong.

## Materials and methods

### Animal specimens

We measured structural brain variation for 63 individuals from 11 gymnotiform species, 7 siluriform species (4 from the electrogenic genus *Synodontis*), 2 characiform species, and 2 cypriniform species. Live cypriniforms, characiforms, siluriforms, and *Eigenmannia virescens* were acquired through the aquarium trade and housed on a 12:12 light:dark cycle in 25–29°C water. Live *Danio rerio* were provided by Dr. Emilia Martins. Formalin-fixed gymnotiform and *S. petricola* specimens were provided by Dr. James Albert and Dr. Jason Gallant, respectively.

## Fixation

Live *Synodontis* and *D. rerio* were euthanized in 600 and 300 mg/mL, respectively, tricaine methanesulfonate (MS-222), immersion fixed in 4% buffered paraformaldehyde for 2 weeks, and then transferred to 70% ethanol. Specimens were decapitated and heads transferred to 0.1 M phosphate buffer prior to scanning, except for *D. rerio* whose small size allowed them to be scanned whole. The remaining live fish were anesthetized in 300 mg/mL MS-222, euthanized by transcardial perfusion with 4% paraformaldehyde, and decapitated following methods in *Sukhum et al., 2018*. These methods are consistent with euthanasia guidelines by the American Veterinary Medical Association and have been approved by the Animal Care and Use Committee at Washington University in St. Louis.

## *Synodontis* electrical recordings

Prior to fixation, we recorded from live *Synodontis* spp. following previous methods (*Boyle et al., 2014*) to determine whether they were electrogenic as the electrogenic abilities of most *Synodontis*, including the species in this study, remain unknown. Briefly, one or two individuals were placed into a tank containing a PVC tube for shelter and a differential recording electrode. We recorded continuously in 2 min intervals for a total of 60 min. Signals were 500× amplified, bandpass filtered (1 Hz to 50 kHz, BMA-200, CWE Inc, Ardmore, PA), and digitized at 48.8 kHz (16-bit PCM converter, RX8, Tucker Davis Technologies, Alachua, FL) using custom MATLAB scripts (*Schumacher and Carlson, 2022*). We recorded electrical discharges from all three tested species (*Figure 2—figure supplement 1*). We were unable to try recording from *S. petricola* to confirm electrogenesis. Lack of recording does not mean that they are incapable of producing electrical discharges, and only 1 out of 13 tested *Synodontis* species did not produce electrical discharges under experimental conditions (*Baron et al., 1994*; *Baron, 2002*; *Boyle et al., 2014*; *Hagedorn et al., 1990*). Further, the confirmed electrogenic *Synodontis* spp. are broadly distributed throughout the species radiation (*Day et al., 2013*; *Pinton et al., 2013*), so additional research is needed to identify the number of origins and losses of electrogenesis among *Synodontis* fishes. Given the apparent lability of electrogenesis in *Synodontis* catfishes, these fishes would be a good place to study the intermediate relationships between evolutionary changes in brain structure and evolution of electrogenesis.

## Micro-computed tomography scans

Heads were contrast stained in 2% phosphomolybdic acid (PMA) for 1 week for small specimens (mass <0.4 g), 5% PMA for 1 week for medium specimens (0.4 g≤ mass < 14 g), or 8% PMA for 2 weeks for large specimens (mass ≥ 14 g) and then transferred to 0.1 M phosphate buffer. μCT scans were done at the Musculoskeletal Research Center at the Barnes-Jewish Institute of Health using a SCANCO μCT40 (Medical model 10 version SCANO_V1.2a, Brüttisellen, Switzerland) following scan conditions in *Sukhum et al., 2018*. Slice thickness ranged from 6 to 18 μm, and scan tube diameter ranged from 12 to 36 mm depending on specimen size.

## Brain region delimitation

We used neuroanatomical landmarks based on previous neuroanatomical studies (*Abrahão et al., 2018*; *Loomis et al., 2019*; *Maler et al., 1991*; *Ullmann et al., 2010*) to consistently delineate brain regions (*Figure 9*). Below, we define planes used to distinguish boundaries between regions in addition to the external and internal surfaces of the brain. We followed the natural breaks in continuous brain tissue wherever possible, which on occasion permitted continuous brain tissue to cross the boundaries set by the planes. We only allowed this when the natural breaks in the brain tissue were obvious and unambiguous, and we never allowed for crossing of the posterior-HB plane (dark blue). The horizontal plane (white) divides the brain into dorsal and ventral areas. It extends from the most ventral point between the telencephalon and optic tectum (landmark a) to the most dorsal bulge of the spinal cord (landmark b).

The olfactory bulb (OB) is a small, ellipsoid bulb at the anterior of the brain connected to the olfactory nerve. In gymnotiforms, characiforms, some cypriniforms, and some siluriforms, the olfactory bulb is the smaller bulb adjacent to the telencephalon. On the posterior side, the olfactory bulb is separated by the olfactory plane (magenta), which starts at the most posterior point of the anterior side of the telencephalon (landmark c) and extends to the most dorsal point underneath the telencephalon (landmark d). On the anterior side, the olfactory bulb is separated from the olfactory tract by

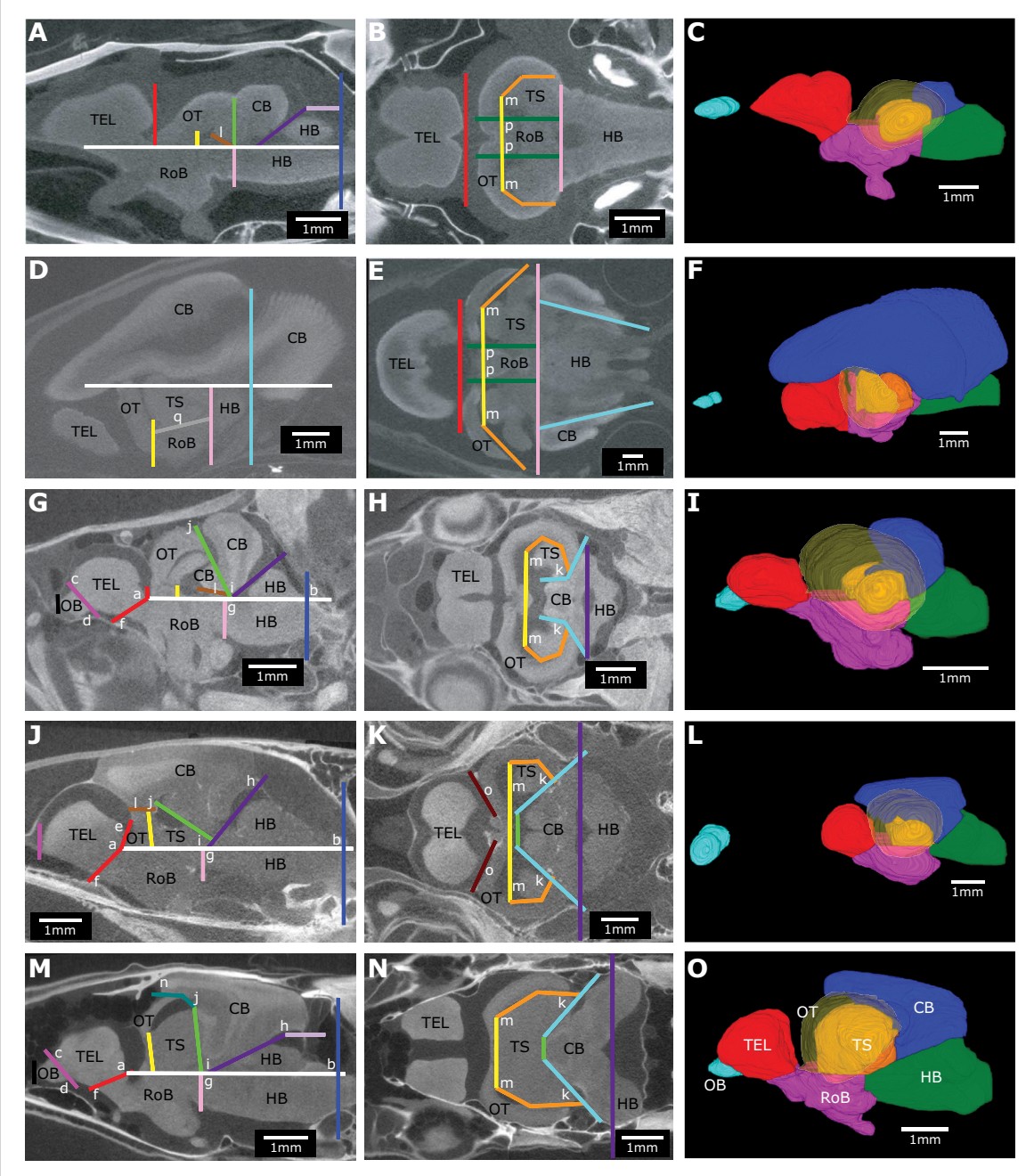

**Figure 9.** Brain landmarks and planes used to consistently delineate brain regions across species. Example brain slices and 3D reconstructions from *Pantodon buchholzi* (**A–C**), *Gnathonemus petersii* (**D–F**), *Phenocogrammus interruptus* (**G–I**), *Synodontis petricola* (**J–L**), and *Eigenmannia virescens* (**M–O**) that show the landmarks (letters) and planes (lines). Osteoglossiform brain slices (**A, B, E**) were modified from *Sukhum et al., 2018*. Brain slices are oriented facing left in a sagittal plane (**A, D, G, J, M**) and horizontal plane (**B, E, H, K, N**). Images were made by averaging across 10 adjacent slices (**A, B, E**) or 5 adjacent slices (**D, G, H, J, K, M, N**). 3D reconstructions have a semi-transparent optic tectum to show the torus semicircularis. Brain regions are color-coded: OB, olfactory bulbs (cyan); TEL, telencephalon (red); HB, hindbrain (green); OT, optic tectum (yellow); TS, torus semicircularis (orange); CB, cerebellum (blue); RoB, rest of brain (magenta). Scale bar = 1 mm.

The online version of this article includes the following source data for figure 9:

**Source data 1.** Coefficient of variation results of repeated measures (N = 3) for four different brains.

a straight plane (black) at the base of the bulge, that is, the olfactory bulb. In the remaining species, the olfactory bulb is in the anterior region of the skull cavity and is clearly separated from the rest of brain by the olfactory nerve.

The telencephalon (TEL) is the larger ellipsoid bulb at the anterior of the brain. There is a clear fissure separating the telencephalon from the more posterior regions of the brain. On the anterior side, the telencephalon is separated by the olfactory plane (magenta). The posterior-TEL plane (red) is a connection of three points: a, the most ventral point between the telencephalon and optic tectum, where it meets the horizontal plane; e, the most posterior bulge of the telencephalon; and f, the lower concave curve of the telencephalon, which is just anterior to the optic nerve. In some cases, a and e are the same point.

The hindbrain (HB) is the most posterior region both above and below the horizontal plane. On the anterior side above the horizontal plane, the plane separating the hindbrain (CB-HB plane, dark purple) extends from the most ventral point between the hindbrain and cerebellum (landmark g) to the concave curve of the hindbrain (landmark h). This most ventral point is a clear cistern separating the cerebellum and hindbrain when viewed in a frontal slice. In species where the cerebellum extends dorsally over the hindbrain, the horizontal-CB plane (light purple) extends from the end of the CB-HB plane (dark purple) parallel to the horizontal plane along the ventral side of the cerebellum. On the anterior side below the horizontal plane, the hindbrain is separated by the anterior-HB plane (light pink) marked by the concave curve of the cerebellum (landmark i) and extending in a straight line perpendicular to the horizontal plane (white). On the posterior side, the hindbrain is separated from the spinal cord by the posterior-HB plane (dark blue): a straight line perpendicular to the horizontal plane that is marked by the most posterior point of the cerebellum or dorsal bulge of the spinal cord (landmark b), whichever is most posterior.

The cerebellum (CB) is the most dorsal region of the brain. It extends from the optic tectum to the hindbrain, but sometimes covers the telencephalon in mormyroids. On the anterior side, the cerebellum is clearly separated from the optic tectum and torus semicircularis. Following this separation, the anterior-CB plane (light green) extends from the top of the optic tectum (landmark j) to the horizontal plane. The lateral-CB plane (cyan) separates the remainder of the torus semicircularis from the cerebellum and connects from the end of the anterior-CB plane and follows the posterior curve of the torus semicircularis to connect to the most posterior concave curve of the torus semicircularis (landmark k). On the posterior side, the cerebellum is separated from the hindbrain by the CB-HB plane (purple). On the ventral side, the cerebellum is separated from the hindbrain and rest of brain by the horizontal plane (white). In *Synodontis*, the optic tectum and torus semicircularis are more lateral and the anterior end of the cerebellum extends further ventral. To separate this part of the cerebellum from the optic tectum, there is an additional ventral-CB plane (brown) extending between the posterior-TEL plane (red) and OT-TS plane (yellow) along the most ventral point at the anterior of the cerebellum (landmark l). In non-electric species, the cerebellum extends anteriorly between the optic tectum and dorsal to the torus semicircularis and rest of brain. To define these anterior boundaries of the cerebellum, the lateral-CB plane (cyan) consists of a second plane that extends further anterior along the lateral sides of the cerebellum. The ventral-CB plane (brown) extends between the most anterior point of the cerebellum to the anterior-CB plane (light green) along the most ventral point at the anterior of the cerebellum (landmark l).

The optic tectum (OT) is the most lateral bulge of the brain and forms a cup-like structure around the torus semicircularis and rest of the midbrain. On the anterior side, the optic tectum is separated from the telencephalon by the posterior-TEL plane (red). On the lateral and posterior sides, the optic tectum is separated from the torus semicircularis by the OT-TS plane (yellow) and the lateral-TS planes (orange). The OT-TS plane (yellow) follows the curve of the torus semicircularis and extends medial-laterally connecting the furthest anterior curves of the torus semicircularis (landmark m). The lateral-TS planes (orange) extend from the end of the OT-TS plane (yellow) to the furthest lateral curve of the torus semicircularis. In gymnotiforms, this requires two planes, but in siluriforms, characiforms, and cypriniforms, this requires three or four planes due to the optic tectum wrapping more tightly around the torus semicircularis. Dorsally, the optic tectum is separated from the cerebellum by the OT-CB plane (teal), which extends from the end of the anterior-CB plane (light green) following along the curve of the optic tectum to the most anterior, concave curve of the cerebellum (landmark n). This requires two planes in some species due to a more anteriorly extended cerebellum. In *Synodontis*,

the optic tectum is more distal to the midline of the brain than in gymnotiforms and instead the OT-CB plane (teal) extends from the most medial and ventral point separating the optic tectum from the cerebellum along the curve of the optic tectum to the anterior-CB plane (light green). In *Synodontis*, there is an additional plane to separate the anterior of the optic tectum from the cerebellum; this anterior-OT plane (dark red) extends from the most anterior curve of optic tectum (landmark o) moving medially along the curve of the optic tectum to the OT-CB plane (teal).

The torus semicircularis (TS) is the two symmetrical, ellipsoid bulbs within the cup of the optic tectum. The torus semicircularis is clearly separated from the more anterior and lateral optic tectum by the OT-TS plane (yellow) and the lateral-TS planes (orange). On the posterior side, the torus semicircularis is clearly separated from the cerebellum by the anterior-CB plane (light green), lateral-CB plane (cyan), and ventral-CB plane (brown). On the ventral side, the torus semicircularis is separated from the rest of brain by the horizontal plane (white). For the osteoglossiform brains, we used the landmarks and planes in *Sukhum et al., 2018* with additional planes to separate the torus semicircularis from the rest of brain. The boundaries of the torus semicircularis in outgroup osteoglossiforms are equivalent to those used for outgroup otophysans with the addition of a medial boundary (optic tectum medial planes, dark green) to separate the torus semicircularis from the rest of brain. The optic tectum medial planes (dark green) extend along the furthest lateral curve of the thalamus (landmark p) as in *Sukhum et al., 2018* but were modified to extend further posterior to intersect the anterior-HB plane (light pink). In mormyroids, the enlarged cerebellum pushes the torus semicircularis further ventral, below the horizontal plane (white). The torus semicircularis is separated from the optic tectum by the OT-TS plane (called optic tectum plane in *Sukhum et al., 2018*, yellow) and the lateral-TS planes (lateral optic tectum planes, orange). The torus semicircularis is separated dorsally from the cerebellum by the horizontal plane (white), posteriorly from the hindbrain by the anterior-HB plane (light pink), and medially from the rest of brain by the optic tectum medial planes (dark green), which were modified to extend further posterior to the anterior-HB plane (light pink). On the ventral side, the torus semicircularis is separated from rest of brain by the ventral-TS plane (gray), which extends from the most ventral point between the torus semicircularis and optic tectum along the most ventral curve of the torus semicircularis (landmark q) to the anterior-HB plane (light pink).

The rest of brain (RoB) combines the remainder of the undifferentiated brain into one region and is between the horizontal plane (white), posterior-TEL plane (red), and anterior-HB plane (light pink).

In *Sukhum et al., 2018*, the authors did not separate torus semicircularis from the rest of brain because the area that they had defined as torus semicircularis also included non-toral regions of the midbrain. Here, we have decided to separate torus semicircularis from the rest of brain despite this because the non-toral regions of the midbrain that are included within the boundaries of our definition of torus semicircularis are comparable across all of our species with the largest non-toral regions included in the torus semicircularis of our non-electrosensory species. This means that although the torus semicircularis volume is overestimated some in all species, the overestimation is larger in our non-electrosensory species than in our electrosensory species, and thus our findings are potentially more conservative than the real differences in regional volumes. Further, the absolute region volumes are not the focus of the study, rather we are concerned with the relative patterns of region volumes across taxa. A consistent overestimation of torus semicircularis volumes does not change these patterns.

## Quantifying region volume

Brain region volumes were measured using the ImageJ Volumest plugin (*Merzin, 2008*; *Schindelin et al., 2012*; *Schneider et al., 2012*). We manually traced each region every 2–10 slices: regions < 2 mm³ were measured every 2 slices, regions 2–4 mm³ were measured every 5 slices or less, and regions > 4 mm³ were measured every 10 slices or less. Volumest then calculates volume using stereological methods with slice thickness ranging from 6 to 18 µm, depending on specimen size, and a 0.1 mm grid width. We randomly selected four scans to be remeasured twice, blind to previous results and species identity. Coefficients of variation for these remeasures were all less than 4% (*Figure 9—source data 1*).

## Phylogenetic analysis: Brain size analyses

To determine where shifts in slope and intercept in brain–body allometries are likely to have occurred across teleosts, we utilized a previously assembled time-calibrated Actinopterygii phylogeny of

11,638 species (*Rabosky et al., 2018*) and combined our data with brain and body mass data from *Tsuboi, 2021*; *Tsuboi et al., 2018* for a total of 1016 ray-finned fishes. The topological relationships of actinopterygians are highly debated (*Hughes et al., 2018*; *Rabosky et al., 2018*; *Rabosky et al., 2013*). For this analysis, we opted for a time-calibrated phylogeny that includes the most species even though the topology differs from studies with fewer species but more genetic data and alternative topology testing (*Hughes et al., 2018*) and from the phylogeny used in subsequent analyses. Throughout this study, we attempted to correct for phylogenetic relatedness to the best of our ability given the well-established difficulty in resolving the topological relationships of ray-finned fishes.

Prior to this analysis, we transformed brain volumes to brain masses by multiplying the volume by the density of fixed brain tissue. To determine the density of fixed brain tissue, brains from six individuals across three species were dissected and weighed immediately after scanning. We then calculated the mean fixed brain tissue density (1.32 $g/cm^3$, SD = 0.19) for these individuals and estimated the corresponding mass for each of our remaining brain volumes using this mean fixed brain tissue density. Where possible, we included data from unsequenced species using sequence data from the species in the same monophyletic genus with the shortest distance to the genus node. Due to inconsistencies in the original source (*Dubois, 1913*), we removed *Carassius carassius*. We also removed 10 species with very short terminal branches that prevented proper parameter optimization in the subsequent analysis (in particular α, defined below). Species pairs containing very short terminal branches were identified manually by inspecting the phylogeny, and we randomly determined which sister species to drop from these species pairs. This resulted in a combined dataset of 870 teleost species across 46 orders that had both brain–body mass and phylogenetic data. All brain mass and body mass data were then log10 transformed.

We used Bayesian reversible-jump bivariate multiregime OUrjMCMC (*Uyeda et al., 2017*) to identify shifts in both intercept and slope of the brain–body allometric relationship. This approach allows shifts to be identified without assuming their location a priori. We ran 10 parallel chains with different starting points of 2 million iterations each, sampling every 100th iteration, and discarded the first 0.3 samples as burn-in. Reversible-jump chains were primed without any birth–death proposals for 10,000 generations, meaning that initial parameter values were randomly drawn from the prior distributions with the number of shifts (k), but not their location, fixed for 10,000 iterations. The output of the last iteration was then used as the starting point for the reversible-jump chain, where the number of shifts was again allowed to vary, to improve model fit. We used the following priors: half-Cauchy distribution (scale = 0.1) for α (the strength of attraction towards an adaptive optimum) and $\sigma^2$ (the change in the trait value over unit time), conditional Poisson distribution (mean = 1% of total branches in the phylogeny, max = 5% of total branches) for k (the total number of shifts), normal distribution β~N (μ = mean (PGLS slope fit for each of the 13 orders with at least 9 species), σ = sd (PGLS slope fit for each of the 13 orders with at least 9 species) rounded up to the next 0.1) for β (slope), and normal distribution θ~N (μ = mean (PGLS intercept fit for the 13 orders with at least 9 species), σ = sd (PGLS intercept fit for the 13 orders with at least 9 species) rounded up to the next 0.1) for θ (intercept). All analyses were conducted using species means, but intraspecific standard error was included in all OUrjMCMC models. For species with only one individual, we used the average intraspecific error across all species. We determined convergence of each run and of parallel chains by inspecting the diagnostic plots, comparing the identified shifts, and using Gelman's R statistic. Chains were then combined to summarize parameter estimates (effective sample sizes > 500) and identify shifts with a posterior probability >0.2.

We then tested the identified shifts in a PGLS framework. For shifts in lineages (i.e., for each grade) containing at least three descendants, we fit a PGLS model allowing both slope and intercept to vary for each grade while allowing the strength of phylogenetic signal to vary using Pagel's lambda ($\lambda$), where 0 indicates no phylogenetic signal and 1 indicates phylogenetic signal consistent with Brownian motion (*Pagel, 1999*). We then performed a phylogenetically corrected ANCOVA with phylogenetically corrected pairwise post-hoc testing with a Bonferroni correction for each grade. We also fit separate PGLS models of all the identified shifts, and in turn, collapsed each grade to its ancestral grade. These models were fit following maximum likelihood and compared using Akaike information criterion (AIC) with a ΔAIC cutoff of 2 (*Burnham and Anderson, 2002*).

To explicitly address whether shifts in the allometric relationship are associated with the evolution of electrosensory phenotypes, we ran OUrjMCMC models with fixed shifts for (1) taxa with ampullary

electroreceptors (i.e., shifts at the branch leading to teleosts where ampullary electroreceptors were lost and at the branches leading to Notopteridae + Mormyroidae, siluriforms, and gymnotiforms where ampullary electroreceptors were gained); (2) taxa with tuberous electroreceptors (i.e., shifts at the branches leading to mormyroids and gymnotiforms); and (3) taxa with electrogenesis (i.e., shifts at the branches leading to mormyroids, gymnotiforms, and *Synodontis*; no brain size data was available for any electrogenic percomorph lineages). Additionally, we ran OUrjMCMC models with a fixed shift at the branch leading to osteoglossiforms following the finding of a different allometric relationship for osteoglossiforms but not the eight other focal orders in *Tsuboi, 2021*, OUrjMCMC models only allowing shifts in intercept but not slope between grades, and OUrjMCMC models fitting a single allometric relationship across all taxa. To perform model selection, we then estimated the marginal likelihood for each model using stepping-stone sampling (*Xie et al., 2011*) with 50 steps and shape parameters of 0.3 and 1 at 500,000 iterations each and computed Bayes factors.

## Brain region analyses

As all of our focal species were not present in the phylogeny used above, we built a Bayesian phylogenetic tree from 6 aligned and concatenated genes (*16s*, *coI*, *cytb*, *rh1*, *rag1*, and *rag2*) of 189 species spanning Anguilliformes to Ostariophysi using Beast v1.10.4 (*Suchard et al., 2018*). We used a birth–death process tree prior and unlinked, relaxed lognormal clock models. We used unlinked substitution models of HKY + I + G for *rh1* and GTR + I + G for all other genes as determined by jModelTest (*Darriba et al., 2012*; *Guindon and Gascuel, 2003*). To reduce the computational burden and improve taxonomic resolution, we constrained the monophyly of each order, gymnotiform families, elopomorpha, ostariophysi, and gymnotiforms + siluriforms as sister to each other following previous studies that used substantially more sequence data and tested alternative hypotheses of teleost topology but did not include all of the species used in this study (*Hughes et al., 2018*; *Rabosky et al., 2013*). We time calibrated the phylogeny using the fossil dates and justifications in *Rabosky et al., 2013*. We performed two independent Bayesian analyses starting from random trees each with a chain length of 150,000,000 sampled every 10,000 generations. We used Tracer v1.7.1 to confirm convergence of parameter values across both analyses and effective sample size values >200 (*Rambaut et al., 2018*). We combined the output of both analyses after discarding the first 15,000,000 states for each run and estimated the maximum clade credibility tree. Note that the relative positions of *Mormyrus tapirus* and *B. brachyistius* have flipped relative to the *cytb* tree in *Sukhum et al., 2018*. To include data from unsequenced species, we used sequence data from the species in the same monophyletic genus with the shortest distance to the genus node. We pruned the tree to only include species with brain measurement data. Given the uncertainty of phylogenetic relationships both among otophysans and within gymnotiforms (*Crampton, 2019*; *Hughes et al., 2018*; *Rabosky et al., 2013*) and our use of relatively few previously sequenced genes, we make no claims that these are the actual phylogenetic relationships between these species. Instead, our purpose in this study was to correct for relatedness as best we could.

We performed a phylogenetic pPCA on species means and applied those rotations to the data from all individuals. For the z-score normalized pPCA, we normalized region volumes, reran the pPCA on species means, and reapplied those rotations to the normalized data from all individuals. *Motani and Schmitz, 2011* implemented a phylogenetic correction, similar in concept to PGLS, to the flexible discriminant analysis framework developed by *Hastie et al., 1994*, known as phylogenetic flexible discriminant analysis (pFDA). pFDA is a two-step process whereby you first determine the optimal degree of phylogenetic signal (lambda) in the form–function relationship by iterating over a range of lambda values to maximize the linear goodness of fit (i.e., minimizing the residual sum of squares) between the discrete grouping variable (here electrosensory phenotype) and the continuous variables (here brain region volumes). This optimal lambda value is then used in the FDA calculation to correct for phylogeny. Our fitted optimal lambda value was 0, which can result both from a lack of phylogenetic signal in the presence of a form–function correlation or a lack of a form–function correlation in the presence of phylogenetic signal (*Motani and Schmitz, 2011*) and likely reflects the convergence of brain morphology with convergent electrosensory phenotypes. R code for optimizing lambda and performing the pFDA was included with *Motani and Schmitz, 2011* and largely based on code from *Hastie et al., 1994*.

We compared PGLS models considering the null hypothesis of body mass and total brain volume predicting PC1–4 values to PGLS models considering that and either and both of the electrosensory phenotypes. Since *G. niloticus* individuals were received by *Sukhum et al., 2018* as fixed, decapitated specimens, their body mass was unknown, thus we removed them from all PC model fits. PC1 was correlated with total brain volume, as expected in allometric relationships (*Klingenberg, 1996*), so we removed total brain volume as a covariate in all PC1 models. All models were fit following maximum likelihood allowing $\lambda$ to vary and compared using small-sample corrected AICc with a $\Delta$AICc cutoff of 2 (*Burnham and Anderson, 2002*). All PGLS fits were determined using species means.

To test for mosaic shifts, we fit PGLS regressions of each brain region volume against total brain volume, against total brain volume–focal region, and against each other region volume for each group. We allowed $\lambda$ to vary for each brain region and tested for significant differences between groups in both slope and intercept using a phylogenetically corrected ANCOVA. For the three electrosensory phenotypes, we performed phylogenetically corrected pairwise post-hoc tests with a Bonferroni correction for each comparison with significant differences.

All phylogenetic analyses were done using R v3.6.2 and the packages bayou, mda, phytools, ape, nlme, MuMIn, and emmeans (*R Core Team, 2019*; *Uyeda et al., 2020*; *Hastie et al., 2020*; *Lenth, 2020*; *Bartoń, 2013*; *Pinheiro et al., 2019*; *Revell, 2012*; *Paradis et al., 2004*).

## Acknowledgements

We thank James Albert for providing additional gymnotiform specimens; Jason Gallant for providing *Synodontis* spp.; Emilia Martins, Piyumika Suriyampola, and Anuradha Bhat for providing wild-caught *D. rerio*; and Jeroen Smaers for assistance with Bayesian modeling.

## Additional information

### Funding

| Funder | Grant reference number | Author |
| --- | --- | --- |
| National Science Foundation | IOS-1755071 | Bruce A Carlson |

The funders had no role in study design, data collection and interpretation, or the decision to submit the work for publication.

### Author contributions

Erika L Schumacher, Conceptualization, Data curation, Formal analysis, Writing - original draft, Writing - review and editing; Bruce A Carlson, Conceptualization, Writing - review and editing

### Author ORCIDs

Erika L Schumacher http://orcid.org/0000-0002-2492-1117
Bruce A Carlson http://orcid.org/0000-0002-2151-0443

### Ethics

The methods in this study are consistent with euthanasia guidelines by the American Veterinary Medical Association and have been approved by the Animal Care and Use Committee at Washington University in St. Louis (Protocol ID 19-0974).

### Decision letter and Author response

Decision letter https://doi.org/10.7554/eLife.74159.sa1
Author response https://doi.org/10.7554/eLife.74159.sa2

## Additional files

### Supplementary files

• Transparent reporting form

- Supplementary file 1. Brain region data from this study. OB, olfactory bulbs; TEL, telencephalon; HB, hindbrain; OT, optic tectum; TS, torus semicircularis; CB, cerebellum; RoB, rest of brain.
- Supplementary file 2. Ray-finned fishes brain and body mass data used in this study.
- Supplementary file 3. Phylopic credits for the silhouettes in *Figures 1 and 3*.

### Data availability

Brain measurement data is located in Supplementary File 1. Brain mass data is located in Supplementary File 2. All analysis code and phylogenetic trees are available in Dryad. The raw micro-computed tomography scans are too large to post (multiple TBs), but are available upon request. To request raw otophysan and/or osteoglossiform scans, contact the corresponding author. We ask that those who want access to the scan data send us an external hard drive, which we will upload all the data to and then return.

The following dataset was generated:

| Author(s) | Year | Dataset title | Dataset URL | Database and Identifier |
|---|---|---|---|---|
| Schumacher EL, Carlson B | 2022 | Data from: Convergent mosaic brain evolution is associated with the evolution of novel electrosensory systems in teleost fishes | https://doi.org/10.5061/dryad.7d7wm37w5 | Dryad Digital Repository, 10.5061/dryad.7d7wm37w5 |

The following previously published datasets were used:

| Author(s) | Year | Dataset title | Dataset URL | Database and Identifier |
|---|---|---|---|---|
| Tsuboi M, van der Bijl W, Kopperud BT, Erritzoe J, Voje KL, Kotrschal A | 2018 | Data from: Brain mass and body mass datasets and phylogenies linked to brain-body allometry and the encephalization of birds and mammals | https://doi.org/10.6084/m9.figshare.6803276.v1 | figshare, 10.6084/m9.figshare.6803276.v1 |

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
