## [Editor Report]

Much of the observed variation in brain region volumes across vertebrates is explained by the scaling relationship of each region with brain size. Nevertheless, mosaic shifts in region volumes independent of overall brain size appear and are thought to reflect selection on particular behavioral traits associated with those brain regions. The work reported here used the independently evolved electrosensory systems of African weakly electric fish, South American weakly electric fish, and weakly electric catfish to show similar enlargement of brain regions, suggesting that selection is repeatedly able to favor brain regions involved in specific behaviors.

---

## [Decision Letter]

**Decision letter after peer review:**

Thank you for submitting your article "Convergent mosaic brain evolution is associated with the evolution of novel electrosensory systems in teleost fishes" for consideration by *eLife*. Your article has been reviewed by 4 peer reviewers, including Catherine Emily Carr as the Reviewing Editor and Reviewer #1, and the evaluation has been overseen by a Reviewing Editor and Ronald Calabrese as the Senior Editor.

Essential revisions:

Your reviewers have reached a consensus that your manuscript. would be suitable for *eLife* if additional analyses were conducted. We have identified 3 major additions to your analyses that should substantially strengthen your conclusions.

1. Identify where and when changes in slope and intercept have occurred along the teleost phylogeny (e.g. Smaers et al. 2021 or Ksepka et al. 2020). It is our opinion that this analysis would be possible with the addition of the Tsuboi data on body and brain size, since 32 species are not enough. We understand that merging the two datasets could be difficult given the different methods of measurement, but there should be enough overlap in the species measured that it can be tested if the two data sets are too different.

2. Use a discriminant analysis to show more formal convergence between the brains of electrogenic fishes.

3. Finally, we suggest a reanalysis of how the cerebellar, toral, alar brainstem hypertrophy impacts the supposed reduction in the volumes of other brain regions. You report significant decreases in electrogenic clades in the size of the olfactory bulb, rest of the brain, and optic tectum. This may be an artifact that results from including the cerebellum and other enlarged areas in the dependent variable.

*Reviewer #1 (Recommendations for the authors):*

The authors achieved their aims, and in general the results support their conclusion, except for a concern about the analysis of significant decreases in other brain areas.

*Reviewer #2 (Recommendations for the authors):*

1) The authors present in all the figures all the individual used but analyses were done with the mean for each species (which is correct). I think that the mean, not individuals should be included in all figures. I think this would help to better see the different groups as the combination of shapes and colors becomes a little crowded. Also, given that many individuals were used for most species, the measuring error could be included in the PGLS. See Tsuboi 2021 for an example of this.

2) In the case of testing for differences in the relative size of the brain I think it would be important to include the additional data mentioned above and perform an analysis similar to that found in Smaers et al., 2021 (Science), where the point in the phylogeny where changes in allometric scaling between brain and body size happen are determined. This would allow to determine if the evolution of electroreceptors or electrogenesis coincides with grade shits in the brain-body allometry. Another important analysis would be to determine where changes in the evolutionary rate of brain and body size occur. This because of the unique allometric scaling brain and body in mormyrids (Tsuboid 2021) mentioned above, is possible that this change can be explained by a change in body size not brain size.

3) I think Figure 4—figure supplement 2 should be included in the main manuscript as I think that it better shows the differences in cerebellum size and other structures between electrogenic and non electrogenic species. Also, the graphs of cerebellum size vs torus semicircular or hindbrain support the co evolution of this structures. Additionally, I don't think that the graph above and below the diagonal are necessary, as they are redundant and only lead to confusion.

4) Has the volumetric measurement been validated in sectioned brains? It seems to me that the borders shown in figure 7 are very rough and it would be important to know at which degree they actually capture the structures that they intent. This seems particularly relevant for the optic tectum and torus semicircularis.

5) I think is important for the authors to report the loadings of the principal component analysis. The strength and variation in the loading of the principal size axis (PC1) have been usually used as a way to evaluate how much a structure has been influenced by mosaic evolution. My guess is that in this case the loadings of the cerebellum in PC1 are not as strong as other structures.

6) Line 192: are non phylogenetic ANCOVAs been used in this case? This should never be the case. I understand that the lack of post-hoc analysis in a phylogenetic contex is frustrating but non-phylogenetically corrected ANCOVAS are not the solution. See https://smaerslab.com/gls-ancova/ for an implementation of a phylogenetic ANCOVA.

7) Why is figure 6 included? Have the electric discharges for these 3 species not been reported before? Even if this is the case, it doesn’t seem necessary as a main figure in the manuscript, especially at the end. Perhaps as a supplementary material would be better.

8) The use of residuals in figure 2 is not ideal in this case as the different slope of mormyrids makes it so that the small mormyrids have small residuals while the larger ones have large ones.

*Reviewer #3 (Recommendations for the authors):*

Line 45: perhaps explain more clearly that mosaic evolution is well accepted for small cell groups but less clear for major brain regions (which may be "carved up" differently in different species to generate mosaic patterns).

Line 54: it would be good to define "ecomorph" or just describe what you mean without using the word. "Ecomorph brain structure" is especially unclear unless you already know…

Line 72: It is unclear in this sentence whether the "to varying degrees" applies to the independent evolution (e.g., some features evolved independently, some not) or to the similarity (the systems evolved independently but are only partially similar).

Figure 1: it might be good to mention that the relative position of Mormyrus tapirus and B. brachyistius has flipped relative to Sukhum et al., 2018.

Figure 3: It might be good to mention somewhere that your PCA analyses yield what others have called "cerebrotypes"; I think Iwaniuk was one of the earliest to use this approach.

Line 153: Is it justified to write "electrosensory cerebellum"? E.g., in the synodontids, do we know that their cerebellum processes electrosensory information? In general, I'd be cautious in assigning any of your studied regions to a particular sensory modality, since you lack the resolution and functional data to do so.

– Lines 197-200: I found these sentences to contain the most important data, but they got lost a bit amongst all the other results. Might it help to construct an additional figure in which you show at which point in the various lineages you hypothesize what kinds of changes in the brain? I could have used such a figure to get a clearer picture of your main results.

Figure 4: I like this figure, though it is busy. It took me a while to understand your complicated legends and inserts, but it all makes sense. As noted in my "public review" it's hard to know how to interpret these data because dramatic increases in the size of the cerebellum would force at least some other regions to appear as if they'd shrunk, but I still like the figure.

– Line 228: given your finding that the allometric slope of the cerebellum increased in a particular lineage, does that set up the cerebellum to become enlarged through "concerted evolution" as species within this lineage increase in absolute brain size. I.e., have you identified a case where a mosaic evolutionary change leads to shifts in brain region proportions through concerted evolution? In short, I would have liked to see you make more of your findings on changes in intercepts vs. slopes; this is a pretty novel aspect of your findings, which warrants additional scrutiny (I think).

Line 298: Why do you think the Synodontids don't show an enlarged hindbrain relative to other catfishes? Could it be that the hindbrain of catfishes is already pretty enlarged because of their vagal and facial lobes, making it difficult for changes in other hindbrain regions to stand out?

– Line 339: Here is a good example of how your failure to appreciate how increases in the proportional size of one region force decreases in others sets you up to make (potentially) unwarranted claims. I bet mormyroids didn't really reduce the size of their telencephalon, relative to the non-electrogenic relatives (I say that in part because I know that the telencephalon of at least some mormyroids is remarkably well differentiated); it's just that their cerebellum is so enlarged that the telencephalon has become proportionately smaller. I think those are important distinctions to make.

– Line 354: Again, the shifts in proportional brain region size need not involve any changes in neuron number or size. They could simply arise from changes in the size of other brain regions.

– Line 409: The point about electrogenic Synodontids not being monophyletic is kind of buried in the text; how does it affect your interpretations/conclusions?

– Line 422: I was very confused initially by your description of how you delineated brain regions. You make it sound as if you determine region volumes only on the basis of the various planes you describe. That would yield very approximate and misleading volumes. Instead, as I gathered from reading the Sorkhum et al., 2018, paper, you use those planes to mark some boundaries between regions, but you actually trace the outlines of brain regions to get your estimates. As far as I can tell, you are using the brain's external and internal surfaces together with your "planes" to delineate the brain regions. This should be clarified right up front (don't wait until lines 537-539 and 555). I know it may seem obvious to you, but I spent quite a bit of time wondering how you could possibly get the images on the right side of Figure 7 from the various planes you illustrate in the rest of the figure. Call me dumb, but I bet I wouldn't be the only one confused :-

*Reviewer #4 (Recommendations for the authors):*

I have a few recommendations for the authors that I think will strengthen the manuscript as a whole.

One of the weaknesses I highlighted was the lack of sufficient background information for non-specialists. Addressing this minor weakness will increase the impact of the work by broadening readership. I think the same can be said of the Discussion. The last few sentence of the last paragraph are good, but I think could be expanded a bit more. It might help if the authors answered some broader questions here. For example, is this a pattern that might be specific to electrosensory systems? Do you expect to see correlated evolution like this outside of sensory systems? I should clarify that I do not think this needs to be extensive, but a broader approach would again improve readership and impact.

It might also be worth trying some alternative, multivariate approaches. Cluster analysis can be quite useful for these kinds of studies, as demonstrated by several comparative neuroanatomical papers, including several on bony fishes. Although I am unaware of a phylogeny-informed cluster analysis algorithm, comparing the cluster analysis with the phylogeny can be informative to determine the extent to which convergent brain forms have evolved. This could help strengthen the authors' case by presenting the data in a slightly more interpretable way than the PCA or scatterplots of brain region allometry. I will emphasize that I do not consider it necessary to use such an approach, but worth trying out to see if it strengthens the conclusions further.

Also, in relation to the statistics, ensure that within the Results you are clear that when significant differences in intercepts were found, there were no differences in slopes. This is not consistent throughout the results and some readers will take issue if they think everything is based solely on intercepts with no testing for different slopes.

I also suggest that the authors check out the BMC Biology paper by Avin et al. (https://link.springer.com/article/10.1186/s12915-021-01024-1). It is a novel approach to testing the concerted vs. mosaic models that Schumacher and Carlson might want to discuss in the context of their findings.

Last, I recommend incorporating supplemental Figure 4 into the manuscript proper. It was not only interesting, it strongly supported many of the authors' conclusions.

---

## [Author Response]

Essential revisions:Your reviewers have reached a consensus that your manuscript. would be suitable for eLife if additional analyses were conducted. We have identified 3 major additions to your analyses that should substantially strengthen your conclusions.1. Identify where and when changes in slope and intercept have occurred along the teleost phylogeny (e.g. Smaers et al. 2021 or Ksepka et al. 2020). It is our opinion that this analysis would be possible with the addition of the Tsuboi data on body and brain size, since 32 species are not enough. We understand that merging the two datasets could be difficult given the different methods of measurement, but there should be enough overlap in the species measured that it can be tested if the two data sets are too different.

We thank the reviewers for this suggestion. Between the Tsuboi data and our data, we had a total of 870 species with brain size, body size, and sequence data. We found that shifts in the brain body allometry are not associated with the evolution of electrosensory phenotypes, but we did identify a shift in both slope and intercept in the branch leading to osteoglossiforms and at two other locations within actinopterygians. Future research is needed to assess why and how osteoglossiforms were able to evolve this shift, but a shifted allometry is not necessary to evolve an electrosensory system nor mosaic shifts in major brain regions. Instead, we find similar shifts in the region-brain allometries of electrosensory species relative to non-electrosensory species despite differences in brain-body scaling between electrosensory taxa. We have made substantial additions to the manuscript with this analysis, but see in particular methods (lines 778-826), results (lines 110-153), discussion (lines 513-528, 530-538, 569-575), and figure 3 and its source data files.

2. Use a discriminant analysis to show more formal convergence between the brains of electrogenic fishes.

We thank the reviewers for this suggestion. We performed a phylogenetic flexible discriminant analysis and found perfect separation between our electrosensory phenotypes, which is consistent with our pPCA results. We added this analysis to the methods (lines 854-866), results (lines 177-186), and as Figure 4 —figure supplements 2 and Figure 4—source data 3.

3. Finally, we suggest a reanalysis of how the cerebellar, toral, alar brainstem hypertrophy impacts the supposed reduction in the volumes of other brain regions. You report significant decreases in electrogenic clades in the size of the olfactory bulb, rest of the brain, and optic tectum. This may be an artifact that results from including the cerebellum and other enlarged areas in the dependent variable.

We thank the reviewers for this perspective and apologize for the lack of clarity regarding this point. We have added additional analyses of the region x region comparisons and moved this figure to the main text (Figure 6 and associated source data files, Figure 7—figure supplements 2-4, Figure 7—source data 3-5). As pointed out by a reviewer below, regressing each region against rest of brain does help to address this point, but also assumes that there have been no evolutionary changes in any of the other regions that make up rest of brain. However, we know this is not true as the lateral hypothalamic regions and preglomerular complex are known to vary substantially across teleosts, and mormyroids are known to have extensive preglomerular complexes (Wulliman 2020, Wulliman and Northcutt 1990). Unfortunately, we do not have enough species to reconstruct ancestral absolute region volumes to address how these observed mosaic shifts correspond to directional changes in absolute region volumes across this phylogenetic scale. However, the reviewers have brought up good points about how mosaic shifts can result from a relative increase in one region, which also appears as a decrease in one or more other regions due to the use of relative measures. However, we argue that this is not necessarily an artifact as this decrease could be reflected in any number of other region volumes, and thus finding relative decreases associated with the evolution of electrogenesis can still be informative regarding relative investment in different brain regions and potential functional trade-offs, even in the absence of changes in absolute volumes. Future studies will be needed to determine whether there are any changes in absolute region volumes, and their direction. We have incorporated substantial changes to the text to better clarify these points throughout, and incorporate the region x region analyses and combine them with the region x total brain volume results to address this point. In particular, see lines 261-295, 307-317, 330-331, 473-499.

Reviewer #1 (Recommendations for the authors):The authors achieved their aims, and in general the results support their conclusion, except for a concern about the analysis of significant decreases in other brain areas.

We thank the reviewer for identifying this miscommunication. We have added additional region x region analyses to help address this point. We have also clarified that all of our identified mosaic shifts are relative to other lineages and do not necessarily reflect changes in absolute region sizes without additional work to directly address evolutionary changes in absolute region sizes. We have incorporated these changes throughout but see in particular lines 261-295, 307-317, 330-331, 473-499 and figure 6. See also our detailed response to essential revision 3.

Reviewer #2 (Recommendations for the authors):1) The authors present in all the figures all the individual used but analyses were done with the mean for each species (which is correct). I think that the mean, not individuals should be included in all figures. I think this would help to better see the different groups as the combination of shapes and colors becomes a little crowded. Also, given that many individuals were used for most species, the measuring error could be included in the PGLS. See Tsuboi 2021 for an example of this.

We thank the reviewer for this suggestion. We agree that some of the plots are too crowded and switched to plotting means instead of individuals for all region by region plots (figure 6, figure 7—figure supplement 2-4). We agree that it is important to incorporate measurement error in analyses where possible, and we did attempt to include this. Unfortunately, we did not have the statistical power to incorporate measurement error in these analyses with our sample size considering there is error in both the x and the y variables as well as in the relationship between them (see Ives et al. 2007). We did include measurement error in all OUrjMCMC analyses with 870 species (lines 794-797).

2) In the case of testing for differences in the relative size of the brain I think it would be important to include the additional data mentioned above and perform an analysis similar to that found in Smaers et al., 2021 (Science), where the point in the phylogeny where changes in allometric scaling between brain and body size happen are determined. This would allow to determine if the evolution of electroreceptors or electrogenesis coincides with grade shits in the brain-body allometry. Another important analysis would be to determine where changes in the evolutionary rate of brain and body size occur. This because of the unique allometric scaling brain and body in mormyrids (Tsuboid 2021) mentioned above, is possible that this change can be explained by a change in body size not brain size.

We thank the reviewer for this suggestion. We directly tested the hypotheses that grade shifts are associated with the evolution of electrosensory phenotypes and found that this is not supported as well as the unconstrained shifts found from fitting across all of the data (lines 143-153, figure 3—source data 4). See also our detailed response to essential revision 1. However, we do not feel comfortable making claims about the relative magnitude of changes in brain size vs body size evolution as done in Smaers et al. 2021 because unlike birds and mammals, fish have indeterminate growth. The brain and body sizes of fish that have been studied do not necessarily reflect the full range of possible adult brain and body sizes for many species. Additionally, there is likely a taxonomic bias in which some species better represent the full range of body sizes due to the ease with which some groups can be collected relative to others, in particular with respect to deep sea and pelagic fishes. For these reasons, we are unable to make any claims about the relationship of brain size evolution to body size evolution.

3) I think Figure 4—figure supplement 2 should be included in the main manuscript as I think that it better shows the differences in cerebellum size and other structures between electrogenic and non electrogenic species. Also, the graphs of cerebellum size vs torus semicircular or hindbrain support the co evolution of this structures. Additionally, I don't think that the graph above and below the diagonal are necessary, as they are redundant and only lead to confusion.

We thank the reviewer for this suggestion. We have moved this figure to the main text (Figure 6) and included discussion of their results. See in particular lines 261-295. However, we disagree that the graphs above and below the diagonal are redundant as which region is the dependent versus independent variable differs, and since we are regressing the regions against each other, this distinction makes a difference.

4) Has the volumetric measurement been validated in sectioned brains? It seems to me that the borders shown in figure 7 are very rough and it would be important to know at which degree they actually capture the structures that they intent. This seems particularly relevant for the optic tectum and torus semicircularis.

We thank the reviewer for this comment. We did not perform any histology on these brains, but we did use previous histological studies to inform our region boundaries. We have updated the text to include this (line 629). There are three methods commonly used to collect region volume measurements in studies like this: histology, ellipsoid modeling, and intact imaging techniques such as micro-computed tomography and micro-magnetic resonance imaging. Unfortunately, they all have flaws and require defining a boundary between regions. A couple studies, including one in fish, have directly compared these three methods and advocate for the use of intact imaging techniques over both histology and ellipsoid modeling as histology is prone to shrinkage, especially in the telencephalon, and many brain regions are clearly not ellipsoids (Ullman et al. 2010, White and Brown 2015). We agree that some of our boundaries are rough and do include some brain tissue from outside of that region, in particular with the torus semicircularis (see discussion of this point in lines 733-744). However, these overestimations, and thus underestimations of other regions (of rest of brain in the case of torus semicircularis), are relatively small and more importantly, consistent across all of the taxa in this study. As we are interested in the relative patterns of region volumes across taxa and not the absolute region volumes, a consistent overestimation of torus semicircularis, or any other region, does not alter these patterns.

5) I think is important for the authors to report the loadings of the principal component analysis. The strength and variation in the loading of the principal size axis (PC1) have been usually used as a way to evaluate how much a structure has been influenced by mosaic evolution. My guess is that in this case the loadings of the cerebellum in PC1 are not as strong as other structures.

We thank the reviewer for this suggestion. We have added the loadings as Figure 4—source data 1. We find that the loadings on PC1 are comparable for all brain regions.

6) Line 192: are non phylogenetic ANCOVAs been used in this case? This should never be the case. I understand that the lack of post-hoc analysis in a phylogenetic contex is frustrating but non-phylogenetically corrected ANCOVAS are not the solution. See https://smaerslab.com/gls-ancova/ for an implementation of a phylogenetic ANCOVA.

We thank the reviewer for identifying this miscommunication. We corrected for phylogeny in each of our ANCOVAs in the same manner as recommended by Smaers and Rohlf (2016, Evolution) and others, and we apologize for the lack of clarity in the results. We have edited the text throughout the methods and results and figure 5 and 7 legends to clarify that all ANCOVAs were phylogenetically corrected. All post-hoc analyses were also performed on the PGLS relationships following Mull et al. 2020. We have edited the text to clarify that all post-hoc analyses have a phylogenetic correction as well.

7) Why is figure 6 included? Have the electric discharges for these 3 species not been reported before? Even if this is the case, it doesn’t seem necessary as a main figure in the manuscript, especially at the end. Perhaps as a supplementary material would be better.8) The use of residuals in figure 2 is not ideal in this case as the different slope of mormyrids makes it so that the small mormyrids have small residuals while the larger ones have large ones.

We thank the reviewer for this suggestion. Unfortunately, synodontids are woefully understudied—only 13 out of 200+ species have been investigated for electrogenesis despite the first finding 30+ years ago. Electrical discharges had not previously been reported from the species in this study, but we have moved this figure from the main text to figure 2—figure supplement 1 as this was not the primary goal of this study.

Reviewer #3 (Recommendations for the authors):

Line 45: perhaps explain more clearly that mosaic evolution is well accepted for small cell groups but less clear for major brain regions (which may be "carved up" differently in different species to generate mosaic patterns).

We thank the reviewer for this suggestion and have better clarified this distinction throughout, but see in particular lines 42-48.

Line 54: it would be good to define "ecomorph" or just describe what you mean without using the word. "Ecomorph brain structure" is especially unclear unless you already know…

We thank the reviewer for identifying this point of confusion. We defined ecomorph in line 58.

Line 72: It is unclear in this sentence whether the "to varying degrees" applies to the independent evolution (e.g., some features evolved independently, some not) or to the similarity (the systems evolved independently but are only partially similar).

We thank the reviewer for identifying this point of confusion. We have clarified that different lineages evolved electrosensory phenotypes independently from mormyroids, and that which lineages have which electrosensory phenotypes varies in line 83.

Figure 1: it might be good to mention that the relative position of Mormyrus tapirus and B. brachyistius has flipped relative to Sukhum et al., 2018.

We thank the reviewer for this comment. We added this to the methods (line 842). We would like to note that in this study we used genetic data from 6 genes while the tree in Sukhum et al. 2018 was made from only one gene.

Figure 3: It might be good to mention somewhere that your PCA analyses yield what others have called "cerebrotypes"; I think Iwaniuk was one of the earliest to use this approach.

We thank the reviewer for this suggestion and have incorporated the use of cerebrotype throughout but in particular with line 182 and figure 4.

Line 153: Is it justified to write "electrosensory cerebellum"? E.g., in the synodontids, do we know that their cerebellum processes electrosensory information? In general, I'd be cautious in assigning any of your studied regions to a particular sensory modality, since you lack the resolution and functional data to do so.

We thank the reviewer for this suggestion. We removed electrosensory associated from line 170.

– Lines 197-200: I found these sentences to contain the most important data, but they got lost a bit amongst all the other results. Might it help to construct an additional figure in which you show at which point in the various lineages you hypothesize what kinds of changes in the brain? I could have used such a figure to get a clearer picture of your main results.

We thank the reviewer for this suggestion. We added an additional figure (figure 8) summarizing our main findings.

Figure 4: I like this figure, though it is busy. It took me a while to understand your complicated legends and inserts, but it all makes sense. As noted in my "public review" it's hard to know how to interpret these data because dramatic increases in the size of the cerebellum would force at least some other regions to appear as if they'd shrunk, but I still like the figure.

We thank the reviewer for this comment. We agree that the figure is complex. However, we argue that these results demonstrate the relative differences in region scaling without necessarily demonstrating how absolute region volumes have changed without further research into more taxa and quantifying neuron numbers. See also our detailed response to essential revision 3.

– Line 228: given your finding that the allometric slope of the cerebellum increased in a particular lineage, does that set up the cerebellum to become enlarged through "concerted evolution" as species within this lineage increase in absolute brain size. I.e., have you identified a case where a mosaic evolutionary change leads to shifts in brain region proportions through concerted evolution? In short, I would have liked to see you make more of your findings on changes in intercepts vs. slopes; this is a pretty novel aspect of your findings, which warrants additional scrutiny (I think).

We thank the reviewer for this comment. We have added further discussion on evolutionary changes in intercept vs changes in slope (lines 543-559), but both are likely contributing to the extreme enlargement of the cerebellum in mormyroids. However, it is not currently possible to identify the order in which these changes occurred without further research into how evolution of developmental and intraspecific allometric relationships contribute to differences in interspecific allometries. Much recent work (e.g. Ksepka et al. 2020, Smaers et al. 2021) has really highlighted the importance of considering differences in slope between lineages.

Line 298: Why do you think the Synodontids don't show an enlarged hindbrain relative to other catfishes? Could it be that the hindbrain of catfishes is already pretty enlarged because of their vagal and facial lobes, making it difficult for changes in other hindbrain regions to stand out?

We thank the reviewer for this comment. We agree that the enlarged facial and vagal lobes could potentially be obscuring a relationship with hindbrain size and electrogenesis in *Synodontis* (lines 384-390). This could also be related to the evolution of tuberous electroreceptors which substantially increase the amount of electrosensory input to the brain relative to ampullary electroreceptors (lines 391-394).

– Line 339: Here is a good example of how your failure to appreciate how increases in the proportional size of one region force decreases in others sets you up to make (potentially) unwarranted claims. I bet mormyroids didn't really reduce the size of their telencephalon, relative to the non-electrogenic relatives (I say that in part because I know that the telencephalon of at least some mormyroids is remarkably well differentiated); it's just that their cerebellum is so enlarged that the telencephalon has become proportionately smaller. I think those are important distinctions to make.

We thank the reviewer for identifying this miscommunication. We have clarified that this relative decrease in the telencephalon of mormyroids is relative to non-electric outgroups, but doesn’t necessarily reflect a change in absolute region size (lines 444-447, 478-487).

– Line 354: Again, the shifts in proportional brain region size need not involve any changes in neuron number or size. They could simply arise from changes in the size of other brain regions.

We thank the reviewer for this comment. We have added this point to the discussion and better clarified the distinction between relative and absolute changes in region scaling (lines 473-489).

Line 409: The point about electrogenic Synodontids not being monophyletic is kind of buried in the text; how does it affect your interpretations/conclusions?

We apologize for the miscommunication and lack of clarity regarding this point. We meant to communicate that of the 200+ species of *Synodontis*, only 13 species total have been tested for electrogenesis. 12 of those species produced electrical discharges under experimental conditions. These 12 species do not come from a single clade and instead are broadly distributed across the *Synodontis* phylogeny. Thus, it remains unknown how electrogenesis has evolved in *Synodontis* with respect to the number of origins and losses. As the three species we were able to test in this study produced electrical discharges, we stand by our finding that mosaic shifts in TS (but to a lesser degree than tuberous receptor fishes) and CB are associated with electrogenesis. However, more research is needed to determine how these regions vary across other *Synodontis* species who may or may not produce electrical discharges. Given that 1 of the tested species did not produce discharges under experimental conditions, we argue that these fishes would be a good model to study the intermediate relationships between evolutionary changes in brain structure and the evolution of electrogenesis. We have clarified this point in lines (612-620) and also added more on electrogenic catfishes in general to the discussion (lines 425-437).

– Line 422: I was very confused initially by your description of how you delineated brain regions. You make it sound as if you determine region volumes only on the basis of the various planes you describe. That would yield very approximate and misleading volumes. Instead, as I gathered from reading the Sorkhum et al., 2018, paper, you use those planes to mark some boundaries between regions, but you actually trace the outlines of brain regions to get your estimates. As far as I can tell, you are using the brain's external and internal surfaces together with your "planes" to delineate the brain regions. This should be clarified right up front (don't wait until lines 537-539 and 555). I know it may seem obvious to you, but I spent quite a bit of time wondering how you could possibly get the images on the right side of Figure 7 from the various planes you illustrate in the rest of the figure. Call me dumb, but I bet I wouldn't be the only one confused :-

We thank the reviewer for identifying this point of confusion. We adjusted the text to clarify up front that we did indeed use the external and internal surfaces of the brain in addition to our defined planes when manually tracing brain regions to estimate their volumes (lines 631-636).

Reviewer #4 (Recommendations for the authors):I have a few recommendations for the authors that I think will strengthen the manuscript as a whole.One of the weaknesses I highlighted was the lack of sufficient background information for non-specialists. Addressing this minor weakness will increase the impact of the work by broadening readership. I think the same can be said of the Discussion. The last few sentence of the last paragraph are good, but I think could be expanded a bit more. It might help if the authors answered some broader questions here. For example, is this a pattern that might be specific to electrosensory systems? Do you expect to see correlated evolution like this outside of sensory systems? I should clarify that I do not think this needs to be extensive, but a broader approach would again improve readership and impact.

We thank the reviewer for this comment. We have added further broadened the discussion to address these points (lines 500-512).

It might also be worth trying some alternative, multivariate approaches. Cluster analysis can be quite useful for these kinds of studies, as demonstrated by several comparative neuroanatomical papers, including several on bony fishes. Although I am unaware of a phylogeny-informed cluster analysis algorithm, comparing the cluster analysis with the phylogeny can be informative to determine the extent to which convergent brain forms have evolved. This could help strengthen the authors' case by presenting the data in a slightly more interpretable way than the PCA or scatterplots of brain region allometry. I will emphasize that I do not consider it necessary to use such an approach, but worth trying out to see if it strengthens the conclusions further.

We thank the reviewer for this suggestion. We added a phylogenetic flexible discriminant analysis and found perfect separation between the electrosensory phenotypes. We added this analysis to the methods (lines 854-866), results (lines 177-186), and as figure 4 —figure supplements 2 and figure 4—source data 3. See also our detailed response to essential revision 2.

Also, in relation to the statistics, ensure that within the Results you are clear that when significant differences in intercepts were found, there were no differences in slopes. This is not consistent throughout the results and some readers will take issue if they think everything is based solely on intercepts with no testing for different slopes.

We thank the reviewer for identifying the miscommunication. We have clarified where slope differences were nonsignificant throughout but see lines 259, 306, 318 in particular.

I also suggest that the authors check out the BMC Biology paper by Avin et al. (https://link.springer.com/article/10.1186/s12915-021-01024-1). It is a novel approach to testing the concerted vs. mosaic models that Schumacher and Carlson might want to discuss in the context of their findings.

We thank the reviewer for recommending this paper. We agree that their findings fit nicely with ours and have added discussion of these points in lines 513-528.

Last, I recommend incorporating supplemental Figure 4 into the manuscript proper. It was not only interesting, it strongly supported many of the authors' conclusions.

We thank the reviewer for this suggestion. We have moved this figure to the main text (figure 6).